# SoteriaFL: A Unified Framework for Private Federated Learning with Communication Compression

**Zhize Li**
Carnegie Mellon University
zhizel@andrew.cmu.edu

**Haoyu Zhao**
Princeton University
haoyu@princeton.edu

**Boyue Li**
Carnegie Mellon University
boyuel@andrew.cmu.edu

**Yuejie Chi**
Carnegie Mellon University
yuejiec@andrew.cmu.edu

## Abstract

To enable large-scale machine learning in bandwidth-hungry environments such as wireless networks, significant progress has been made recently in designing communication-efficient federated learning algorithms with the aid of communication compression. On the other end, privacy-preserving, especially at the client level, is another important desideratum that has not been addressed simultaneously in the presence of advanced communication compression techniques yet. In this paper, we propose a unified framework that enhances the communication efficiency of private federated learning with communication compression. Exploiting both general compression operators and local differential privacy, we first examine a simple algorithm that applies compression directly to differentially-private stochastic gradient descent, and identify its limitations. We then propose a unified framework SoteriaFL for private federated learning, which accommodates a general family of local gradient estimators including popular stochastic variance-reduced gradient methods and the state-of-the-art shifted compression scheme. We provide a comprehensive characterization of its performance trade-offs in terms of privacy, utility, and communication complexity, where SoteriaFL is shown to achieve better communication complexity without sacrificing privacy nor utility than other private federated learning algorithms without communication compression.

## 1 Introduction

With the proliferation of mobile and edge devices, federated learning (FL) [42, 55] has recently emerged as a disruptive paradigm for training large-scale machine learning models over a vast amount of geographically distributed and heterogeneous devices. For instance, Google uses FL in the Gboard mobile keyboard for next word predictions [29]. FL is often modeled as a distributed optimization problem [41, 42, 55, 35, 72], aiming to solve

$$\min_{\boldsymbol{x} \in \mathbb{R}^d} \left\{ f(\boldsymbol{x}; D) := \frac{1}{n} \sum_{i=1}^{n} f(\boldsymbol{x}; D_i) \right\}, \text{ where } f(\boldsymbol{x}; D_i) := \frac{1}{m} \sum_{j=1}^{m} f(\boldsymbol{x}; d_{i,j}). \tag{1}$$

Here, $D$ denotes the entire dataset distributed across all $n$ clients, where each client $i$ has a local dataset $D_i = \{d_{i,j}\}_{j=1}^{m}$ of equal size $m$.[1] $\boldsymbol{x} \in \mathbb{R}^d$ denotes the model parameters, $f(\boldsymbol{x}; D)$, $f(\boldsymbol{x}; D_i)$, and $f(\boldsymbol{x}; d_{i,j})$ denote the nonconvex loss function of the current model $\boldsymbol{x}$ on the entire dataset $D$, the local dataset $D_i$, and a single data sample $d_{i,j}$, respectively. For simplicity, we use $f(\boldsymbol{x})$, $f_i(\boldsymbol{x})$ and $f_{i,j}(\boldsymbol{x})$ to denote $f(\boldsymbol{x}; D)$, $f(\boldsymbol{x}; D_i)$ and $f(\boldsymbol{x}; d_{i,j})$, respectively.

---

[1]This is without loss of generality, since otherwise one can simply adjust the weights of the loss function.

36th Conference on Neural Information Processing Systems (NeurIPS 2022).

## 1.1 Motivation: privacy-utility-communication trade-offs

To unleash the full potential of FL, it is extremely important that the algorithm designed to solve (1) needs to meet several competing desiderata.

**Communication efficiency.** Communication between the server and clients is well recognized as the main bottleneck for optimizing the latency of FL systems, especially when the clients—such as mobile devices—have limited bandwidth, the number of clients is large, and/or the machine learning model has a lot of parameters—for example, the language model GPT-3 [7] has billions of parameters and therefore cumbersome to share directly.

Therefore, it is very important to design FL algorithms to reduce the overall communication cost, which takes into account both *the number of communication rounds* and *the cost per communication round* for reaching a desired accuracy. With these two quantities in mind, there are two principal approaches for communication-efficient FL: 1) *local methods*, where in each communication round, clients run multiple local update steps before communicating with the server, in the hope of reducing the number of communication rounds, e.g., [55, 48, 39, 27, 38, 71, 60, 9, 47, 46, 2, 78, 58, 57]; 2) *compression methods*, where clients send compressed communication message to the server, in the hope of reducing the cost per communication round, e.g., [4, 40, 70, 31, 37, 56, 61, 52, 28, 51, 62, 21, 45, 77, 79, 63]. While both categories have garnered significant attention in recent years, we focus on the second approach based on communication compression to enhance communication efficiency.

**Privacy preserving.** While FL holds great promise of harnessing the inferential power of private data stored on a large number of distributed clients, these local data at clients often contain sensitive or proprietary information without consent to share. Although FL may appear to protect the data privacy via storing data locally and only sharing the model updates (e.g., gradient information), the training process can nonetheless reveal sensitive information as demonstrated by, e.g., Zhu et al. [81]. It is thus desirable for FL to preserve privacy in a guaranteed manner [24, 35, 64, 72].

To ensure the training process does not accidentally leak private information, advanced privacy-preserving tools such as *differential privacy* (DP) [20] have been widely integrated into training algorithms [18, 12, 19, 1, 69, 32, 15, 23]. A notable example is Abadi et al. [1], which developed a differentially-private stochastic gradient descent (SGD) algorithm DP-SGD in the centralized (single-node) setting. More recently, several differentially-private algorithms [33, 73, 65, 54] are proposed for the more general distributed ($n$-node) setting suitable for FL. In this paper, we also follow the DP approach to preserve privacy. In particular, we adopt local differential privacy (LDP) to respect the privacy of each client, which is critical in FL.

**Goal.** Encouraged by recent advances in communication compression techniques, and the widespread success of differentially-private methods, a natural question is

*Can we develop a unified framework for private federated learning with communication compression, and understand the trade-offs between privacy, utility, and communication?*

Note that there have been a handful of works that simultaneously address compression and privacy in FL. Unfortunately, they only provide partial answers to the above question. Most of the existing works only consider specific, elementary, or tailored compression schemes that are applied directly to the gradient messages in DP-SGD [3, 74, 26, 82, 76, 17]. A number of works [66, 13, 14, 36, 22, 67] extended and considered different compression schemes, but did not provide concrete trade-offs in terms of privacy, utility and communication. Furthermore, existing theoretical analyses can be limited only to convex problems [26], lacking in some aspects such as utility [82], or delivering pessimistic guarantees on utility and/or communication due to strong assumptions [76, 17]. Finally, existing work only studied the DP framework for direct compression, while it is known that the recently developed shifted compression scheme [56, 30, 50] achieves much better convergence guarantees. Due to noise injection for privacy-preserving, it is a priori unclear if the shifted compression scheme is also compatible with privacy.

## 1.2 Our contributions

In this paper, we answer the above question by providing a general approach that enhances the communication efficiency of private federated learning in the *nonconvex* setting, through a unified framework called SoteriaFL (see Algorithm 2). Specifically, we have the following contributions.

Table 1: Comparisons among (local) differentially-private algorithms for the nonconvex problem (1) in both central (single-node) and distributed ($n$-node) settings. Here, $m$ denotes the number of data stored on a single client, $n$ is the number of clients, $d$ is the dimension, and $\omega$ is the parameter for the compression operator (cf. Definition 1). The communication complexity is computed by $ndT/(1+\omega)$, where $T$ is the total number of communication rounds, and $nd/(1+\omega)$ is the communication cost per round. The utility / accuracy measures the average squared gradient norm of the objective function after $T$ rounds. Note that the algorithm is better when the utility/accuracy and the communication complexity are small under the same privacy guarantee.

| Algorithm | Privacy | Utility/Accuracy | Communication Complexity | Remark |
|---|---|---|---|---|
| RRPSGD [75] | $(\epsilon,\delta)$-DP | $\frac{\sqrt{d\log(m/\delta)\log(1/\delta)}}{m\epsilon}$ | — | single node |
| DP-GD/SGD [1, 69] | $(\epsilon,\delta)$-DP | $\frac{\sqrt{d\log(1/\delta)}}{m\epsilon}$ | — | single node |
| DP-SRM [73] | $(\epsilon,\delta)$-DP | $\frac{\sqrt{d\log(1/\delta)}}{m\epsilon}$ | — | single node |
| Distributed DP-SRM [73] (1) | $(\epsilon,\delta)$-DP | $\frac{\sqrt{d\log(1/\delta)}}{nm\epsilon}$ | $\frac{n^2 m\epsilon\sqrt{d}}{\sqrt{\log(1/\delta)}}$ | $n$ nodes, no comp. |
| LDP SVRG LDP SPIDER [54] | $(\epsilon,\delta)$-LDP | $\frac{\sqrt{d\log(1/\delta)}}{\sqrt{n}m\epsilon}$ | $\frac{n^{3/2}m\epsilon\sqrt{d}}{\sqrt{\log(1/\delta)}}$ | $n$ nodes, no comp. |
| Q-DPSGD-1 [17] (2) | $(\epsilon,\delta)$-LDP | $\frac{(\tilde{\sigma}^2/n+1/m)^{2/3}(d\log(1/\delta))^{1/3}}{m^{2/3}\epsilon^{2/3}}$ | $\frac{(1+n/(m\tilde{\sigma}^2))m^2\epsilon^2}{d\log(1/\delta)}$ | $n$ nodes, direct comp. |
| SDM-DSGD [76] (3) | $(\epsilon,\delta)$-LDP | $\tilde{O}\left(\frac{\sqrt{d\log(1/\delta)}}{\sqrt{n}m\epsilon}\right)$ | $\frac{n^{7/2}m\epsilon\sqrt{d}}{(1+\omega)^{3/2}\sqrt{\log(1/\delta)}}+\frac{nm^2\epsilon^2}{(1+\omega)\log(1/\delta)}$ | $n$ nodes, direct comp. |
| CDP-SGD (Theorem 1) | $(\epsilon,\delta)$-LDP | $\frac{\sqrt{(1+\omega)d\log(1/\delta)}}{\sqrt{n}m\epsilon}$ | $\frac{n^{3/2}m\epsilon\sqrt{d}}{(1+\omega)^{3/2}\sqrt{\log(1/\delta)}}+\frac{nm^2\epsilon^2}{(1+\omega)\log(1/\delta)}$ | $n$ nodes, direct comp. |
| SoteriaFL-SGD SoteriaFL-GD (4) (Corollary 1) | $(\epsilon,\delta)$-LDP | $\frac{\sqrt{(1+\omega)d\log(1/\delta)}}{\sqrt{n}m\epsilon}(1+\sqrt{\tau})$ | $\frac{n^{3/2}m\epsilon\sqrt{d}}{(1+\omega)^{3/2}\sqrt{\log(1/\delta)}}(1+\sqrt{\tau})$ | $n$ nodes, shifted comp. |
| SoteriaFL-SVRG SoteriaFL-SAGA (4) (Corollary 2, 3) | $(\epsilon,\delta)$-LDP | $\frac{\sqrt{(1+\omega)d\log(1/\delta)}}{\sqrt{n}m\epsilon}$ | $\frac{n^{3/2}m\epsilon\sqrt{d}}{(1+\omega)^{3/2}\sqrt{\log(1/\delta)}}(1+\tau)$ | $n$ nodes, shifted comp. |

(1) Wang et al. [73] considered the "global" $(\epsilon,\delta)$-DP (which only protects the privacy for entire dataset $D$, i.e., the local dataset $D_i$ on node $i$ may leak to other nodes $j\neq i$) without communication compression. However, we consider the "local" $(\epsilon,\delta)$-LDP which can protect the local datasets $D_i$'s at the client level.

(2) Ding et al. [17] adopted a slightly different compression assumption $\mathbb{E}[\|\mathcal{C}(\boldsymbol{x})-\boldsymbol{x}\|^2]\leq\tilde{\sigma}^2$, with $\tilde{\sigma}^2$ playing a similar role as $(1+\omega)$ in ours. However, it obtains a worse accuracy $\frac{(\tilde{\sigma}^2/n+1/m)^{2/3}(d\log(1/\delta))^{1/3}}{m^{2/3}\epsilon^{2/3}}=\frac{\sqrt{(\tilde{\sigma}^2/n+1/m)d\log(1/\delta)}}{m\epsilon}\cdot\left(\frac{(\tilde{\sigma}^2/n+1/m)m^2\epsilon^2}{d\log(1/\delta)}\right)^{1/6}=\frac{\sqrt{(\tilde{\sigma}^2/n+1/m)d\log(1/\delta)}}{m\epsilon}\cdot T^{1/6}$, a factor of $T^{1/6}$ worse than the utility of the other algorithms including ours, where $T=\frac{(\tilde{\sigma}^2/n+1/m)m^2\epsilon^2}{d\log(1/\delta)}$ is the optimal choice to achieve the best accuracy for Q-DPSGD-1.

(3) Zhang et al. [76] only considered random-$k$ sparsification, which is a special case of our general compression operator. Moreover, it requires $1+\omega\ll\log T$, i.e., at least $k\gg\frac{d}{\log T}$ out of $d$ coordinates need to be communicated, and its utility hides logarithmic factors larger than $1+\omega$. The communication complexity $n^{7/2}$ is due to their convergence condition $T>n^5$.

(4) Here, $\tau:=\frac{(1+\omega)^{3/2}}{n^{1/2}}$. If $n\geq(1+\omega)^3$ (which is typical in FL), then $\tau<1$, and we can drop the terms involving $\tau$ from SoteriaFL.

1. We first present a simple algorithm CDP-SGD (Algorithm 1) that directly combines communication compression and DP-SGD. We provide theoretical analysis for CDP-SGD in Theorem 1 and show its limitations in communication efficiency.

2. We then propose a general framework SoteriaFL for private FL, which accommodates a general family of local gradient estimators including popular stochastic variance-reduced gradient methods and the state-of-the-art shifted compression scheme. We provide a unified characterization of its performance trade-offs in terms of privacy, utility (convergence accuracy), and communication complexity.

3. We apply our unified analysis for SoteriaFL and obtain theoretical guarantees for several new private FL algorithms, including SoteriaFL-GD, SoteriaFL-SGD, SoteriaFL-SVRG, and SoteriaFL-SAGA. All of these algorithms are shown to perform better than the plain CDP-SGD (Algorithm 1), and have lower communication complexity compared with other

private FL algorithms without compression. The numerical experiments also corroborate the theory and confirm the practical superiority of SoteriaFL.

We provide detailed comparisons between the proposed approach and prior arts in Table 1. To the best of our knowledge, SoteriaFL is the first unified framework that simultaneously enables local differential privacy and shifted compression, and allows flexible local computation protocols at the client level.

## 2 Preliminaries

Let $[n]$ denote the set $\{1, 2, \cdots, n\}$ and $\|\cdot\|$ denote the Euclidean norm of a vector. Let $\langle \boldsymbol{u}, \boldsymbol{v} \rangle$ denote the standard Euclidean inner product of two vectors $\boldsymbol{u}$ and $\boldsymbol{v}$. Let $f^* := \min_{\boldsymbol{x}} f(\boldsymbol{x}) > -\infty$ denote the optimal value of the objective function in (1). In addition, we use the standard order notation $O(\cdot)$ to hide absolute constants. We now introduce the definitions of the compression operator and local differential privacy, as well as some standard assumptions for the objective functions.

**Compression operator.** Let us introduce the notion of a randomized *compression operator*, which is used to compress the gradients to save communication. The following definition of unbiased compressors is standard and has been used in many distributed/federated learning algorithms [4, 40, 56, 30, 50, 52, 28, 51].

**Definition 1** (Compression operator). *A randomized map* $\mathcal{C} : \mathbb{R}^d \mapsto \mathbb{R}^d$ *is an* $\omega$-*compression operator if for all* $\boldsymbol{x} \in \mathbb{R}^d$, *it satisfies*

$$\mathbb{E}[\mathcal{C}(\boldsymbol{x})] = \boldsymbol{x}, \qquad \mathbb{E}\left[\|\mathcal{C}(\boldsymbol{x}) - \boldsymbol{x}\|^2\right] \le \omega \|\boldsymbol{x}\|^2 . \tag{2}$$

*In particular, no compression* $(\mathcal{C}(\boldsymbol{x}) \equiv \boldsymbol{x})$ *implies* $\omega = 0$.

Note that the conditions (2) are satisfied by many practically useful compression operators, e.g., random sparsification and random quantization [4, 52, 51]. A useful rule of thumb is that the communication cost is often reduced by a factor of $\frac{1}{1+\omega}$ due to compression [4]. Next, we briefly discuss an example called random sparsification to provide more intuition.

**Example 1** (Random sparsification). Given $\boldsymbol{x} \in \mathbb{R}^d$, the random-$k$ sparsification operator is defined by $\mathcal{C}(\boldsymbol{x}) := \frac{d}{k} \cdot (\boldsymbol{\xi}_k \odot \boldsymbol{x})$, where $\odot$ denotes the Hadamard (element-wise) product and $\boldsymbol{\xi}_k \in \{0, 1\}^d$ is a uniformly random binary vector with $k$ nonzero entries ($\|\boldsymbol{\xi}_k\|_0 = k$). This random-$k$ sparsification operator $\mathcal{C}$ satisfies (2) with $\omega = \frac{d}{k} - 1$, and the communication cost is reduced by a factor of $\frac{1}{1+\omega}$ since we transmit $k = \frac{d}{1+\omega}$ (due to $\omega = \frac{d}{k} - 1$) coordinates rather than $d$ coordinates of the message.

**Local differential privacy.** We not only want to train the machine learning model using fewer communication bits, but also want to maintain each client's local privacy, which is a key component for FL applications. Following the framework of (local) differential privacy [5, 11, 80], we say that two datasets $D$ and $D'$ are neighbors if they differ by only one entry. We have the following definition for local differential privacy (LDP).

**Definition 2** (Local differential privacy (LDP)). *A randomized mechanism* $\mathcal{M} : \mathcal{D} \to \mathcal{R}$ *with domain* $\mathcal{D}$ *and range* $\mathcal{R}$ *is* $(\epsilon, \delta)$-*locally differentially private for client* $i$ *if for all neighboring datasets* $D_i, D_i' \in \mathcal{D}$ *on client* $i$ *and for all events* $S \in \mathcal{R}$ *in the output space of* $\mathcal{M}$, *we have*

$$\Pr\{\mathcal{M}(D_i) \in S\} \le e^\epsilon \Pr\{\mathcal{M}(D_i') \in S\} + \delta.$$

The definition of LDP (Definition 2) is very similar to the original definition of $(\epsilon, \delta)$-DP [20, 19], except that now in the FL setting, each client protects its own privacy by encoding and processing its sensitive data locally, and then transmitting the encoded information to the server without coordination and information sharing between the clients.

**Assumptions about the functions.** Recalling (1), we consider the *nonconvex* FL setting, where the functions $\{f_{i,j}\}$ are arbitrary functions satisfying the following standard smoothness assumption (Assumption 1) and bounded gradient assumption (Assumption 2).

**Assumption 1** (Smoothness). *There exists some* $L \ge 0$, *such that for all* $i \in [n], j \in [m]$, *the function* $f_{i,j}$ *is* $L$-*smooth, i.e.,*

$$\|\nabla f_{i,j}(\boldsymbol{x}_1) - \nabla f_{i,j}(\boldsymbol{x}_2)\| \le L \|\boldsymbol{x}_1 - \boldsymbol{x}_2\|, \qquad \forall \boldsymbol{x}_1, \boldsymbol{x}_2 \in \mathbb{R}^d.$$

**Assumption 2** (Bounded gradient)**.** *There exists some $G \geq 0$, such that for all $i \in [n], j \in [m]$ and $\boldsymbol{x} \in \mathbb{R}^d$, we have $\|\nabla f_{i,j}(\boldsymbol{x})\| \leq G$.*

The smoothness assumption is very standard for the convergence analysis [59, 25, 53, 49], and the bounded gradient assumption is also standard for the differential privacy analysis [6, 69, 32, 23].

## 3 Warm-up: Plain Compressed Differentially-Private SGD

There are two methods to combine privacy and compression: (1) first perturb and then compress, and (2) first compress and then perturb. The advantage of the first method is that it is very simple and general, since compression will preserve the differential privacy and work seamlessly with any existing privacy mechanisms. However, the second method requires carefully designed perturbation mechanisms (otherwise the perturbation might diminish the communication saving of compression), e.g., binomial perturbation [3] or discrete Gaussian perturbation [36]. In addition, it is observed that the first method achieves better utility compared with the second one in some settings [17]. Thus, we also apply the first method in this paper: first perturb then compress.

**Baseline algorithm: CDP-SGD.** As a warm-up, we first introduce a simple algorithm CDP-SGD (described in Algorithm 1), which subsumes some existing algorithms as special cases (e.g., [76, 82]) for private FL with better theoretical guarantees. The procedure for CDP-SGD is very simple: at round $t$, each client $i$ first computes a local stochastic gradient $\tilde{\boldsymbol{g}}_i^t$ using its local dataset $D_i$ (Line 4 in Algorithm 1). Then, it uses Gaussian mechanism [1] to achieve LDP (Line 5 in Algorithm 1) and communicates the compressed perturbed private gradient information to the server (Line 6 in Algorithm 1). Finally, the server aggregates the compressed information and update the model parameters (Line 8–9 in Algorithm 1).

---

**Algorithm 1** Compressed Differentially-Private Stochastic Gradient Descent (CDP-SGD)

---

**Input:** initial point $x^0$, stepsize $\eta_t$, variance $\sigma_p^2$, minibatch size $b$
1: **for** $t = 0, 1, 2, \dots, T$ **do**
2:     **for each node** $i \in [n]$ **do in parallel**
3:         Sample a random minibatch $\mathcal{I}_b$ from local dataset $D_i$
4:         Compute local stochastic gradient $\tilde{\boldsymbol{g}}_i^t = \frac{1}{b} \sum_{j \in \mathcal{I}_b} \nabla f_{i,j}(\boldsymbol{x}^t)$    // all nodes use SGD method
5:         *Privacy*: $\boldsymbol{g}_i^t = \tilde{\boldsymbol{g}}_i^t + \boldsymbol{\xi}_i^t$, where $\boldsymbol{\xi}_t^i \sim \mathcal{N}(\boldsymbol{0}, \sigma_p^2 \boldsymbol{I})$
6:         *Compression*: let $\boldsymbol{v}_i^t = \mathcal{C}_i^t(\boldsymbol{g}_i^t)$ and send to the server    // direct compression
7:     **end each node**
8:     Server aggregates compressed information $\boldsymbol{v}^t = \frac{1}{n} \sum_{i=1}^{n} \boldsymbol{v}_i^t$
9:     $\boldsymbol{x}^{t+1} = \boldsymbol{x}^t - \eta_t \boldsymbol{v}^t$
10: **end for**

---

Now we present the theoretical guarantees for CDP-SGD in the following theorem.

**Theorem 1** (Privacy, utility and communication for CDP-SGD)**.** *Suppose that Assumptions 1 and 2 hold, and the compression operators $\mathcal{C}_i^t$ (cf. Line 6 of Algorithm 1) are drawn independently satisfying Definition 1. By choosing the algorithm parameters properly and letting the total number of communication rounds $T = O\left( \frac{\sqrt{n}Lm\epsilon}{G\sqrt{(1+\omega)d\log(1/\delta)}} + \frac{m^2\epsilon^2}{d\log(1/\delta)} \right)$, CDP-SGD (Algorithm 1) satisfies $(\epsilon, \delta)$-LDP and the utility $\frac{1}{T} \sum_{t=0}^{T-1} \mathbb{E}\|\nabla f(\boldsymbol{x}_t)\|^2 \leq O\left( \frac{G\sqrt{(1+\omega)Ld\log(1/\delta)}}{\sqrt{n}m\epsilon} \right).$*

The proposed CDP-SGD (Algorithm 1) is simple but effective. When the compression parameter $\omega$ is a constant (i.e., constant compression ratio), CDP-SGD achieves the same utility $O\left( \frac{\sqrt{d\log(1/\delta)}}{m\epsilon} \right)$ as DP-SGD in the single-node case with $n = 1$. In comparison, our utility is better than [17] by a factor of $T^{1/6}$, and our communication complexity is much better than [76] (see Table 1).

However, the communication complexity of CDP-SGD still has room for improvements due to *direct compression* (Line 6 in Algorithm 1). In particular, if the size of the local dataset $m$ stored on clients is dominating, then CDP-SGD (even if we compute local full gradients as CDP-GD) requires $O(m^2)$ communication rounds (see Theorem 1), while previous distributed differentially-private

algorithms without communication compression (e.g., Distributed DP-SRM [73], LDP SVRG and LDP SPIDER [54]) only need $O(m)$ communication rounds (see Table 1).

# 4 SoteriaFL: A Unified Private FL Framework with Shifted Compression

Due to the limitations of plain CDP-SGD, we now present an advanced and unified private FL framework called SoteriaFL in this section, which allows a large family of local gradient estimators (Line 3 in Algorithm 2 and Line 3–11 in Algorithm 3). Via adopting the advanced *shifted compression* (Line 5 in Algorithm 2), SoteriaFL reduces the total number of communication rounds $O(m^2)$ of CDP-SGD to $O(m)$, which matches previous uncompressed DP algorithms (see Table 1), and further reduces the total communication complexity due to less communication cost per round.

## 4.1 A unified SoteriaFL framework

Our SoteriaFL framework is described in Algorithm 2. At round $t$, each client will compute a local (stochastic) gradient estimator $\tilde{g}_i^t$ using its local dataset $D_i$ (Line 3 in Algorithm 2). One can choose several optimization methods for computing this local gradient estimator such as standard gradient descent (GD), stochastic GD (SGD), stochastic variance reduced gradient (SVRG) [34, 43], and SAGA [16] (see e.g., Line 3–11 in Algorithm 3). Then, each client adds a Gaussian perturbation $\xi_i^t$ on its gradient estimate $\tilde{g}_i^t$ to ensure LDP (Line 4 in Algorithm 2). However, different from CDP-SGD (Algorithm 1) where we directly compress the perturbed stochastic gradients, now each client maintains a reference $s_i^t$ and compresses the shifted message $\tilde{g}_i^t - s_i^t$ (Line 5 in Algorithm 2). This extra shift operation achieves much better convergence behavior (fewer communication rounds) than CDP-SGD, and thus allowing much lower communication complexity.

---

**Algorithm 2** SoteriaFL (a unified framework for compressed private FL)

---

**Input:** initial point $x^0$, stepsize $\eta_t$, shift stepsize $\gamma_t$, variance $\sigma_p^2$, initial reference $s_i^0 = 0$
1: **for** $t = 0, 1, 2, \ldots, T$ **do**
2:      **for each node** $i \in [n]$ **do in parallel**
3:          Compute local gradient estimator $\tilde{g}_i^t$      // it allows many methods, e.g., SGD, SVRG, and SAGA
4:          *Privacy*: $g_i^t = \tilde{g}_i^t + \xi_i^t$, where $\xi_i^t \sim \mathcal{N}(0, \sigma_p^2 I)$
5:          *Compression*: let $v_i^t = \mathcal{C}_i^t(g_i^t - s_i^t)$ and send to the server      // shifted compression
6:          Update shift $s_i^{t+1} = s_i^t + \gamma_t \mathcal{C}_i^t(g_i^t - s_i^t)$
7:      **end each node**
8:      Server aggregates compressed information $v^t = s^t + \frac{1}{n} \sum_{i=1}^n v_i^t$
9:      $x^{t+1} = x^t - \eta_t v^t$
10:     $s^{t+1} = s^t + \gamma_t \frac{1}{n} \sum_{i=1}^n v_i^t$
11: **end for**

---

## 4.2 Generic assumption and unified theory

We provide a generic Assumption 3, which is very flexible to capture the behavior of several existing (and potentially new) gradient estimators, while simultaneously maintaining the tractability to enable a unified and sharp theoretical analysis.

**Assumption 3** (Generic assumption of local gradient estimator for SoteriaFL). *The gradient estimator $\tilde{g}_i^t$ (Line 3 of Algorithm 2) is unbiased $\mathbb{E}_t[\tilde{g}_i^t] = \nabla f_i(x^t)$ for $i \in [n]$, where $\mathbb{E}_t$ takes the expectation conditioned on all history before round $t$. Moreover, it can be decomposed into two terms $\tilde{g}_i^t := \mathcal{A}_i^t + \mathcal{B}_i^t$ and there exist constants $G_A, G_B, C_1, C_2, C_3, C_4, \theta$ and a random sequence $\{\Delta^t\}$ such that*

$$\mathcal{A}_i^t = \frac{1}{b} \sum_{j \in \mathcal{I}_b} \varphi_{i,j}^t, \qquad \mathcal{B}_i^t = \frac{1}{m} \sum_{j=1}^m \psi_{i,j}^t, \tag{3a}$$

$$\mathbb{E}_t \Big[ \frac{1}{n} \sum_{i=1}^n \| \tilde{g}_i^t - \nabla f_i(x^t) \|^2 \Big] \leq C_1 \Delta^t + C_2, \tag{3b}$$

$$\mathbb{E}_t \big[ \Delta^{t+1} \big] \leq (1 - \theta) \Delta^t + C_3 \| \nabla f(x^t) \|^2 + C_4 \mathbb{E}_t \| x^{t+1} - x^t \|^2, \tag{3c}$$

where $\varphi_{i,j}^t$ and $\psi_{i,j}^t$ are bounded by $G_A$ and $G_B$ respectively, and $\mathcal{I}_b$ usually denotes a random minibatch with size $b$. Here, $\varphi_{i,j}^t$ and $\psi_{i,j}^t$ should be viewed as functions related to the $j$-th sample $d_{i,j}$ stored on client $i$.

A few comments are in order. Concretely, the decomposition (3a) is used for our unified privacy analysis (i.e., Theorem 2). We can let one of them be $\mathbf{0}$ if the gradient estimator only contains one term or is not decomposable. The parameters $C_1$ and $C_2$ in (3b) capture the variance of the gradient estimators, e.g., $C_1 = C_2 = 0$ if the client computes local full gradient $\tilde{\boldsymbol{g}}_i^t = \nabla f_i(\boldsymbol{x}^t)$, and $C_1 \neq 0$ (note that $\Delta^t$ will shrink in (3c)) and $C_2 = 0$ if the client uses variance-reduced gradient estimators such as SVRG/SAGA. Finally, the parameters $\theta, C_3$ and $C_4$ in (3c) capture the shrinking behavior of the variance (incurred by the gradient estimators), where different variance-reduced gradient methods usually have different shrinking behaviors. More concrete examples to follow in Lemma 1 in Section 5.

**Unified theory for privacy-utility-communication trade-offs.** Given our generic Assumption 3, we can obtain a unified analysis for SoteriaFL framework. The following Theorem 2 unifies the privacy analysis and Theorem 3 unifies the utility and communication complexity analysis.

**Theorem 2** (Privacy for SoteriaFL). *Suppose that Assumption 3 holds. There exist constants $c$ and $c'$, for any $\epsilon < c'b^2T/m^2$ and $\delta \in (0,1)$, SoteriaFL (Algorithm 2) is $(\epsilon, \delta)$-LDP if we choose*

$$\sigma_p^2 = c\frac{(G_A^2/4 + G_B^2)T\log(1/\delta)}{m^2\epsilon^2}. \tag{4}$$

**Theorem 3** (Utility and communication for SoteriaFL). *Suppose that Assumptions 1 and 3 hold, and the compression operators $\mathcal{C}_i^t$ (cf. Line 5 of Algorithm 2) are drawn independently satisfying Definition 1. Set the stepsize as*

$$\eta_t \equiv \eta \leq \min\left\{\frac{1}{(1 + 2\alpha C_4 + 4\beta(1+\omega) + 2\alpha C_3/\eta^2)L}, \frac{\sqrt{\beta n}}{\sqrt{1 + 2\alpha C_4 + 4\beta(1+\omega)}(1+\omega)L}\right\},$$

*where $\alpha = \frac{3\beta C_1}{2(1+\omega)\theta L^2}$, $\forall\beta > 0$, the shift stepsize as $\gamma_t \equiv \sqrt{\frac{1+2\omega}{2(1+\omega)^3}}$, and the privacy variance $\sigma_p^2$ according to Theorem 2. Then, SoteriaFL (Algorithm 2) satisfies $(\epsilon, \delta)$-LDP and the following*

$$\frac{1}{T}\sum_{t=0}^{T-1}\mathbb{E}\|\nabla f(\boldsymbol{x}^t)\|^2 \leq \frac{2\Phi_0}{\eta T} + \frac{3\beta}{(1+\omega)L\eta}\left(C_2 + \frac{c(G_A^2/4 + G_B^2)dT\log(1/\delta)}{m^2\epsilon^2}\right),$$

*where $\Phi_0 := f(\boldsymbol{x}^0) - f^* + \alpha L\Delta^0 + \frac{\beta}{Ln}\sum_{i=1}^n\|\nabla f_i(\boldsymbol{x}^0) - \boldsymbol{s}_i^0\|^2$. By further choosing the total number of communication rounds $T$ as*

$$T = \max\left\{\frac{m\epsilon\sqrt{2(1+\omega)L\Phi_0}}{\sqrt{3\beta cd(G_A^2/4 + G_B^2)\log(1/\delta)}}, \frac{C_2m^2\epsilon^2}{cd(G_A^2/4 + G_B^2)\log(1/\delta)}\right\}, \tag{5}$$

SoteriaFL *has the following utility (accuracy) guarantee:*

$$\frac{1}{T}\sum_{t=0}^{T-1}\mathbb{E}\|\nabla f(\boldsymbol{x}^t)\|^2 \leq O\left(\max\left\{\frac{\sqrt{\beta d(G_A^2/4 + G_B^2)\log(1/\delta)}}{\eta m\epsilon\sqrt{(1+\omega)L}}, \frac{\beta C_2}{(1+\omega)L\eta}\right\}\right). \tag{6}$$

Theorem 3 is a unified theorem for our SoteriaFL framework, which covers a large family of local stochastic gradient methods under the generic Assumption 3. In the next Section 5, we will show that many popular local gradient estimators (GD, SGD, SVRG, and SAGA) satisfy Assumption 3, and thus can be captured by our unified analysis.

## 5 Some Algorithms within SoteriaFL Framework

In this section, we propose several new private FL algorithms (SoteriaFL-GD, SoteriaFL-SGD, SoteriaFL-SVRG and SoteriaFL-SAGA) captured by our SoteriaFL framework. We give a detailed Algorithm 3 which describes all these four SoteriaFL-type algorithms in a nutshell.

To analyze Algorithm 3 using our unified SoteriaFL framework, we begin by showing that these local gradient estimators (GD, SGD, SVRG, and SAGA) satisfy Assumption 3 in the following main lemma, detailing the corresponding parameter values (i.e., $G_A, G_B, C_1, C_2, C_3, C_4$, and $\theta$).

---

**Algorithm 3** SoteriaFL-SGD, SoteriaFL-SVRG, and SoteriaFL-SAGA

---

**Input:** initial point $\boldsymbol{x}^0$, stepsize $\eta_t$, shift stepsize $\gamma_t$, variance $\sigma_p^2$, minibatch size $b$, initial reference $\boldsymbol{s}_i^0 = 0$, initial $\boldsymbol{w}^0 = \boldsymbol{x}^0$ for SVRG or $\boldsymbol{w}_{i,j}^0 = \boldsymbol{x}^0$ for SAGA, probability $p$

1: **for** $t = 0, 1, 2, \ldots, T$ **do**
2:     **for each node** $i \in [n]$ **do in parallel**
3:         Option I: SGD
4:             Compute local SGD estimator $\tilde{\boldsymbol{g}}_i^t = \frac{1}{b} \sum_{j \in \mathcal{I}_b} \nabla f_{i,j}(\boldsymbol{x}^t)$      // GD if choose $b = m$
5:         Option II: SVRG
6:             Compute local SVRG estimator $\tilde{\boldsymbol{g}}_i^t = \frac{1}{b} \sum_{j \in \mathcal{I}_b} (\nabla f_{i,j}(\boldsymbol{x}^t) - \nabla f_{i,j}(\boldsymbol{w}^t)) + \nabla f_i(\boldsymbol{w}^t)$
7:             Update SVRG snapshot point $\boldsymbol{w}^{t+1} = \begin{cases} \boldsymbol{x}^t, & \text{with probability } p \\ \boldsymbol{w}^t, & \text{with probability } 1 - p \end{cases}$
8:         Option III: SAGA
9:             Compute local SAGA estimator:
                $\tilde{\boldsymbol{g}}_i^t = \frac{1}{b} \sum_{j \in \mathcal{I}_b} (\nabla f_{i,j}(\boldsymbol{x}^t) - \nabla f_{i,j}(\boldsymbol{w}_{i,j}^t)) + \frac{1}{m} \sum_{j=1}^m \nabla f_{i,j}(\boldsymbol{w}_{i,j}^t)$
10:             Update SAGA variables $\boldsymbol{w}_{i,j}^{t+1} = \begin{cases} \boldsymbol{x}^t, & \text{for } j \in \mathcal{I}_b \\ \boldsymbol{w}_{i,j}^t, & \text{for } j \notin \mathcal{I}_b \end{cases}$
11:         End Options
12:         *Privacy*: $\boldsymbol{g}_i^t = \tilde{\boldsymbol{g}}_i^t + \boldsymbol{\xi}_i^t$, where $\boldsymbol{\xi}_i^t \sim \mathcal{N}(\boldsymbol{0}, \sigma_p^2 \boldsymbol{I})$
13:         *Compression*: let $\boldsymbol{v}_i^t = \mathcal{C}_i^t(\boldsymbol{g}_i^t - \boldsymbol{s}_i^t)$ and send to the server
14:         Update shift $\boldsymbol{s}_i^{t+1} = \boldsymbol{s}_i^t + \gamma_t \mathcal{C}_i^t(\boldsymbol{g}_i^t - \boldsymbol{s}_i^t)$
15:     **end each node**
16:     Server aggregates compressed information $\boldsymbol{v}^t = \boldsymbol{s}^t + \frac{1}{n} \sum_{i=1}^n \boldsymbol{v}_i^t$
17:     $\boldsymbol{x}^{t+1} = \boldsymbol{x}^t - \eta_t \boldsymbol{v}^t$
18:     $\boldsymbol{s}^{t+1} = \boldsymbol{s}^t + \gamma_t \frac{1}{n} \sum_{i=1}^n \boldsymbol{v}_i^t$
19: **end for**

---

**Lemma 1** (SGD/SVRG/SAGA estimators satisfy Assumption 3)**.** *Suppose that Assumptions 1 and 2 hold. The local SGD estimator $\tilde{\boldsymbol{g}}_i^t$ (Option I in Algorithm 3) satisfies Assumption 3 with*

$$G_A = G, \ G_B = C_1 = C_3 = C_4 = 0, \ C_2 = \frac{(m-b)G^2}{mb}, \ \theta = 1, \ \Delta^t \equiv 0.$$

*The local SVRG estimator $\tilde{\boldsymbol{g}}_i^t$ (Option II in Algorithm 3) satisfies Assumption 3 with*

$$G_A = 2G, \ G_B = G, \ C_1 = \frac{L^2}{b}, \ C_2 = 0, \ C_3 = \frac{2(1-p)\eta^2}{p}, \ C_4 = 1, \ \theta = \frac{p}{2}, \ \Delta^t = \|\boldsymbol{x}^t - \boldsymbol{w}^t\|^2.$$

*The local SAGA estimator $\tilde{\boldsymbol{g}}_i^t$ (Option III in Algorithm 3) satisfies Assumption 3 with*

$$G_A = 2G, \ G_B = G, \ C_1 = \frac{L^2}{b}, \ C_2 = 0, \ C_3 = \frac{2(m-b)\eta^2}{b}, \ C_4 = 1,$$

$$\theta = \frac{b}{2m}, \ \Delta^t = \frac{1}{nm} \sum_{i=1}^n \sum_{j=1}^m \|\boldsymbol{x}^t - \boldsymbol{w}_{i,j}^t\|^2.$$

With Lemma 1 in hand, we can plug their corresponding parameters into the unified Theorem 3 to obtain detailed utility and communication bounds for the resulting methods (SoteriaFL-SGD/SoteriaFL-GD, SoteriaFL-SVRG, and SoteriaFL-SAGA). Formally, we have the following three corollaries.

**Corollary 1** (SoteriaFL-SGD/SoteriaFL-GD)**.** *Suppose that Assumptions 1 and 2 hold and we combine Theorem 3 and Lemma 1, i.e., choosing stepsize $\eta_t \equiv \eta \leq \frac{1}{(1+2\sqrt{(1+\omega)^3/n})L}$, where we set $\beta = \frac{\tau}{2(1+\omega)}$ and $\tau := \frac{(1+\omega)^{3/2}}{n^{1/2}}$, shift stepsize $\gamma_t \equiv \sqrt{\frac{1+2\omega}{2(1+\omega)^3}}$, and privacy variance $\sigma_p^2 = O\left(\frac{G^2 T \log(1/\delta)}{m^2 \epsilon^2}\right)$. If we further set the minibatch size $b = \min\left\{\frac{m\epsilon G\sqrt{\beta}}{\sqrt{(1+\omega)Ld\log(1/\delta)}}, m\right\}$ and the total number of communication rounds $T = O\left(\frac{\sqrt{n}Lm\epsilon}{G\sqrt{(1+\omega)d\log(1/\delta)}}(1 + \sqrt{\tau})\right)$, then*

Table 2: Gradient complexity for our proposed SoteriaFL-style algorithms, which is computed as the product of the total number of communication rounds $T$ and the minibatch size $b$. Here, for notation simplicity, $K := \frac{\sqrt{n}Lm\epsilon}{G\sqrt{(1+\omega)d\log(1/\delta)}}$ and $\tau := \frac{(1+\omega)^{3/2}}{n^{1/2}}$.

| **Algorithms** | SoteriaFL-GD (Option I in Algorithm 3 with $b = m$) | SoteriaFL-SGD (Option I in Algorithm 3) | SoteriaFL-SVRG SoteriaFL-SAGA (Option II, III in Algorithm 3) |
|---|---|---|---|
| **Gradient Complexity** | $K(1 + \sqrt{\tau})m$ | $K(1 + \sqrt{\tau})b$ | $K(1 + \tau)m^{2/3}$ |

SoteriaFL-SGD *satisfies $(\epsilon, \delta)$-LDP and the following utility guarantee* $\frac{1}{T}\sum_{t=0}^{T-1}\mathbb{E}\|\nabla f(\boldsymbol{x}_t)\|^2 \leq O\Big(\frac{G\sqrt{(1+\omega)Ld\log(1/\delta)}}{\sqrt{n}m\epsilon}(1 + \sqrt{\tau})\Big)$. *If we choose a minibatch size $b = m$ (local full gradient) in* SoteriaFL-SGD, *the result of* SoteriaFL-SGD *leads to that of* SoteriaFL-GD.

**Corollary 2** (SoteriaFL-SVRG). *Suppose that Assumptions 1 and 2 hold and we combine Theorem 3 and Lemma 1, i.e., choosing stepsize $\eta_t \equiv \eta \leq \frac{p^{2/3}b^{1/3}\min\{1,\sqrt{n/(1+\omega)^3}\}}{2L}$, where we set $\beta = \frac{p^{4/3}b^{2/3}(1+\omega)^2\min\{1,n/(1+\omega)^3\}}{n}$, $p^{2/3}b^{1/3} \leq 1/4$ and $p \leq 1/4$, shift stepsize $\gamma_t \equiv \sqrt{\frac{1+2\omega}{2(1+\omega)^3}}$, and privacy variance $\sigma_p^2 = O\big(\frac{G^2T\log(1/\delta)}{m^2\epsilon^2}\big)$. If we further let the minibatch size $b = \frac{m^{2/3}}{4}$, the probability $p = b/m$, and the total number of communication rounds $T = O\Big(\frac{\sqrt{n}Lm\epsilon}{G\sqrt{(1+\omega)d\log(1/\delta)}}\max\{1,\tau\}\Big)$, where $\tau := \frac{(1+\omega)^{3/2}}{n^{1/2}}$, then* SoteriaFL-SVRG *satisfies $(\epsilon, \delta)$-LDP and the following utility guarantee* $\frac{1}{T}\sum_{t=0}^{T-1}\mathbb{E}\|\nabla f(\boldsymbol{x}^t)\|^2 \leq O\Big(\frac{G\sqrt{(1+\omega)Ld\log(1/\delta)}}{\sqrt{n}m\epsilon}\Big)$.

The utility and communication complexity for SoteriaFL-SAGA are the same as SoteriaFL-SVRG, and we defer its detailed corollary to the appendix.

Interestingly, SoteriaFL-style algorithms are more communication-efficient than CDP-SGD when the local dataset size $m$ is large, with a communication complexity of $O(m)$, in contrast to $O(m^2)$ for CDP-SGD. In terms of utility, SoteriaFL-SVRG and SoteriaFL-SAGA can achieve the same utility as CDP-SGD, while SoteriaFL-GD and SoteriaFL-SGD achieve a slightly worse guarantee than that of CDP-SGD by a factor of $1 + \sqrt{\tau}$, where $\tau := \frac{(1+\omega)^{3/2}}{n^{1/2}}$ is small when the number of clients $n$ is large.

**Gradient complexity of SoteriaFL-style algorithms.** Although the utility and the communication complexity are the most important considerations in private FL, another worth-noting criterion is the *gradient complexity*, which is defined as the total number of stochastic gradients computed by each client. Although SoteriaFL-GD, SoteriaFL-SGD, SoteriaFL-SVRG and SoteriaFL-SAGA have similar communication complexity (see Table 1), they actually have very different gradient complexities—summarized in Table 2—since the minibatch sizes and gradient update rules for these algorithms vary a lot. The gradient complexity of SoteriaFL-SVRG/SoteriaFL-SAGA is usually smaller than SoteriaFL-SGD, and all of them are smaller than SoteriaFL-GD. In sum, we recommend SoteriaFL-SVRG/SoteriaFL-SAGA due to its superior utility and gradient complexity while maintaining almost the same communication complexity as SoteriaFL-SGD/SoteriaFL-GD.

## 6 Numerical Experiments

We conduct experiments on standard real-world datasets [10, 44] to numerically verify privacy-utility-communication trade-offs among different algorithms. The code can be accessed at:

https://github.com/haoyuzhao123/soteriafl.

Concretely, we compare the direct compression algorithm CDP-SGD (Algorithm 1), shifted compression algorithms SoteriaFL-SGD (Algorithm 3 with Option I) and SoteriaFL-SVRG (Algorithm 3

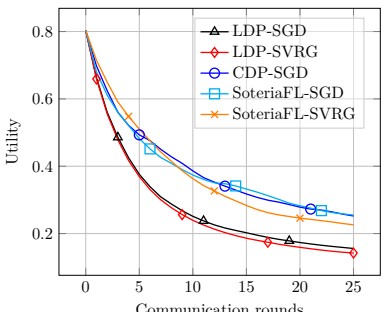
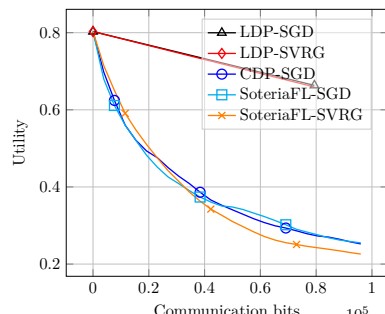

Figure 1: Logistic regression with nonconvex regularization on the `a9a` dataset under $(\epsilon, \delta)$-LDP with $\epsilon = 1$ and $\delta = 10^{-3}$.

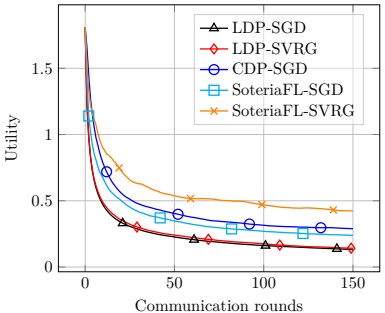
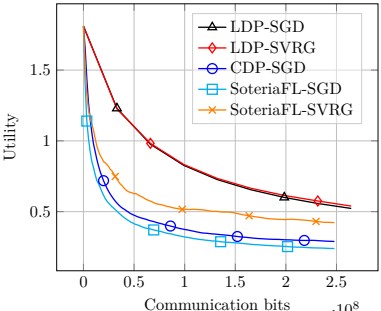

Figure 2: Shallow neural network training on the `MNIST` dataset under $(\epsilon, \delta)$-LDP with $\epsilon = 1$ and $\delta = 10^{-3}$.

with Option II), and algorithms without compression LDP-SGD [1, 54] and LDP-SVRG [54] on two nonconvex problems (logistic regression with nonconvex regularization, and shallow neural network training). The detailed problem definition, experiment setup, and more experiments can be found in Appendix A.

The experimental results show that compressed algorithms converges faster than the uncompressed algorithm in terms of *communication bits* (right columns), and also confirm that shifted compression based SoteriaFL can perform better than direct compression based CDP-SGD.

## 7 Conclusion

We propose SoteriaFL, a unified framework for private FL, which accommodates a general family of local gradient estimators including popular stochastic variance-reduced gradient methods and the state-of-the-art shifted compression scheme. A unified characterization of its performance trade-offs in terms of privacy, utility (convergence accuracy), and communication complexity is presented, which is then instantiated to arrive at several new private FL algorithms. All of these algorithms are shown to perform better than the plain CDP-SGD algorithm especially when the local dataset size is large, and have lower communication complexity compared with other private FL algorithms without compression.

## Acknowledgments

The work of Z. Li, B. Li and Y. Chi is supported in part by ONR N00014-19-1-2404, by AFRL under FA8750-20-2-0504, and by NSF under CCF-1901199, CCF-2007911, DMS-2134080 and CNS-2148212. The work of H. Zhao is supported in part by NSF, ONR, Simons Foundation, DARPA and SRC through awards to S. Arora. B. Li is also gratefully supported by Wei Shen and Xuehong Zhang Presidential Fellowship at Carnegie Mellon University.

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
