# Appendix

We present more experiments and provide all missing proofs in the appendix. Concretely, Appendix A describes the experiment setup and contains additional numerical experiments. Appendix B and C provide the detailed proofs for our unified privacy guarantee in Theorem 2 and unified utility and communication complexity analysis in Theorem 3, respectively. Appendix D provides the proof for CDP-SGD (Theorem 1). Finally, Appendix E provides the proofs for Section 5, including Lemma 1 (showing that several local gradient estimators satisfy the generic Assumption 3) and Corollaries 1–3 (instantiating Lemma 1 in the unified Theorem 3) for the proposed SoteriaFL-style algorithms.

## A Experiments

We describe the problem definition and provide more experiments for logistic regression with nonconvex regularization in Appendix A.1 and shallow neural network training in Appendix A.2.

**Experiment setup.** In our experiments, we use random-$k$ sparsification (see Example 1 in Section 2) as the compression operator, and we set $k = \lfloor \frac{d}{20} \rfloor$, i.e., randomly select 5% coordinates over $d$ dimension to communicate. In other words, the number of communication bits *per round* of uncompressed algorithms equals to that of *20 rounds* of compressed algorithms. The number of nodes $n$ is 10. For the algorithmic parameters, we tune the stepsizes (learning rates) for all algorithms for each nonconvex problem and select their best ones from the set $\{0.01, 0.03, 0.06, 0.1, 0.3, 0.6, 1\}$. Other parameters are set according to their theoretical values. We would like point out that, in order to achieve privacy guarantee, bounded gradient (Assumption 2) is required. However, it is not easy to obtain this upper bound $G$ or it is somewhat large especially for neural networks. Thus, following experiments in previous works [73, 76, 17, 54], we also apply gradient clipping (i.e. $\text{clip}_G(\boldsymbol{g}) = \min(1, \frac{G}{\|\boldsymbol{g}\|}) \cdot \boldsymbol{g}$) in our experiments. In particular, we choose $G = 0.5$ for logistic regression with nonconvex regularization in Appendix A.1 and $G = 1$ for shallow neural network training in Appendix A.2. For the Gaussian perturbation $\boldsymbol{\xi}$, we will run experiments for different levels of $(\epsilon, \delta)$-LDP guarantee, and compute the variance of $\boldsymbol{\xi}$ according to the theory.

### A.1 Logistic regression with nonconvex regularization

The first task is the logistic regression with a nonconvex regularizer, where the objective function over a data sample $(\boldsymbol{a}, b) \in D$ is defined as

$$f(\boldsymbol{x}; (\boldsymbol{a}, b)) := \log\left(1 + \exp(-b\boldsymbol{a}^\top \boldsymbol{x})\right) + \lambda \sum_{j=1}^{d} \frac{x_j^2}{1 + x_j^2}.$$

Here, $\boldsymbol{a} \in \mathbb{R}^d$ denotes the features, $b$ is its label, and $\lambda$ is the regularization parameter. We choose $\lambda = 0.2$ and run the experiments on the standard `a9a` dataset [10]. To demonstrate the privacy-utility-communication trade-offs, we consider three levels of $(\epsilon, \delta)$-LDP with different $\epsilon = 1, 5, 10$ and a common $\delta = 10^{-3}$, where the experimental results are reported in Figure 3.

**Remark.** From the experimental results in Figure 3, it can be seen that the two uncompressed algorithms (LDP-SGD and LDP-SVRG) converge faster than the three compressed algorithms (CDP-SGD, SoteriaFL-SGD, SoteriaFL-SVRG) in terms of *communication rounds* (see left columns). However, in terms of *communication bits* (see right columns), compressed algorithms perform better than the uncompressed algorithms. This validates that communication compression indeed provide significant savings in terms of communication cost. The figure also confirms that shifted compression based SoteriaFL typically performs better than direct compression based CDP-SGD. For SoteriaFL-style algorithms, it turns out that SoteriaFL-SVRG performs slightly better than SoteriaFL-SGD in the utility. This is quite consistent with our theoretical results.

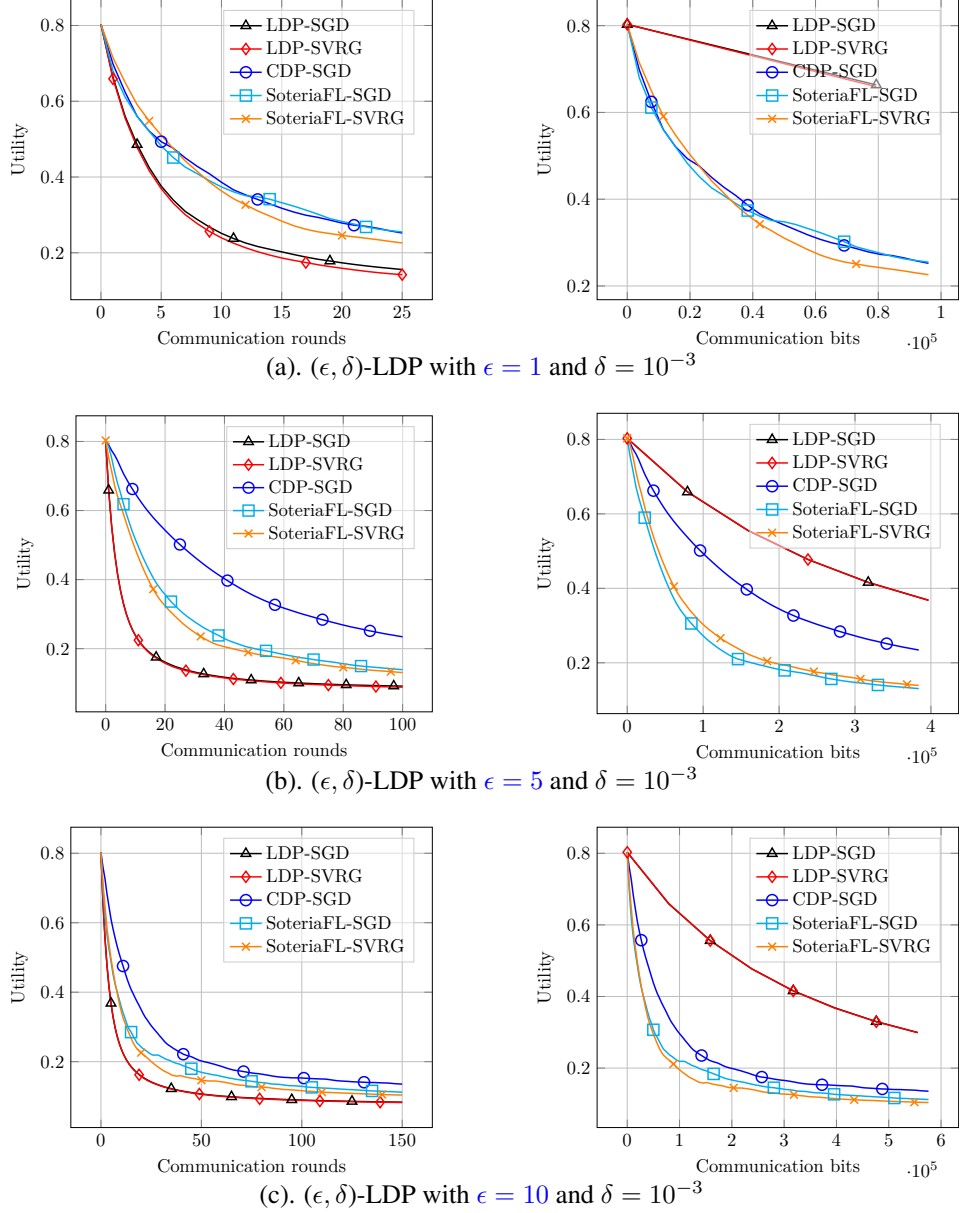

Figure 3: Logistic regression with nonconvex regularization on the a9a dataset under $(\epsilon, \delta)$-LDP with $\epsilon = 1, 5, 10$ and $\delta = 10^{-3}$. The left (resp. right) column is for utility vs. communication rounds (resp. communication bits).

## A.2 Shallow neural network training

We consider a simple 1-hidden layer neural network training task, with $64$ hidden neurons, sigmoid activation functions, and the cross-entropy loss. The objective function over a data sample $(\boldsymbol{a}, b)$ is defined as

$$f(\boldsymbol{x}; (\boldsymbol{a}, b)) = \ell(\mathsf{softmax}(\boldsymbol{W}_2 \, \mathsf{sigmoid}(\boldsymbol{W}_1 \boldsymbol{a} + \boldsymbol{c}_1) + \boldsymbol{c}_2), b),$$

where $\ell(\cdot, \cdot)$ denotes the cross-entropy loss, the optimization variable is collectively denoted by $\boldsymbol{x} = \mathsf{vec}(\boldsymbol{W}_1, \boldsymbol{c}_1, \boldsymbol{W}_2, \boldsymbol{c}_2)$, with the dimensions of the network parameters $\boldsymbol{W}_1, \boldsymbol{c}_1, \boldsymbol{W}_2, \boldsymbol{c}_2$ being $64 \times 784$, $64 \times 1$, $10 \times 64$, and $10 \times 1$, respectively. Here, we run the experiments on the standard MNIST dataset [44]. To demonstrate the privacy-utility-communication trade-offs, we consider five levels of $(\epsilon, \delta)$-LDP with $\epsilon = 1, 2, 4, 8, 16$ and a common $\delta = 10^{-3}$, where the experimental results are reported in Figures 4–8, respectively.

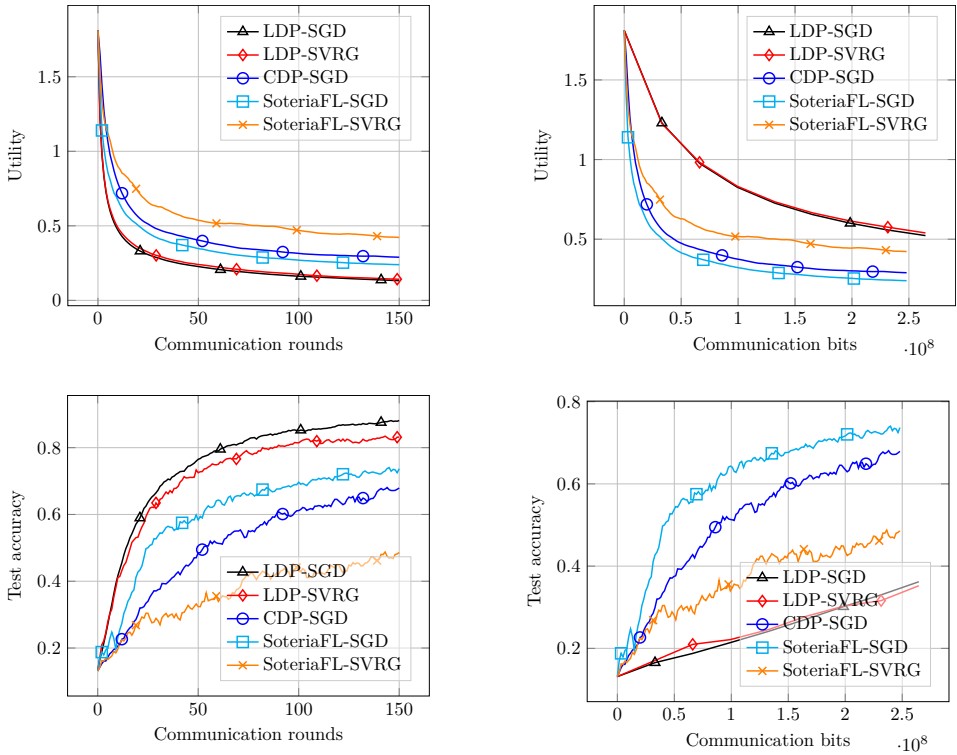

Figure 4: Shallow neural network training on the MNIST dataset under $(\epsilon, \delta)$-LDP with $\epsilon = 1$ and $\delta = 10^{-3}$. The top (resp. bottom) row is for utility (resp. test accuracy) vs. communication rounds and communication bits.

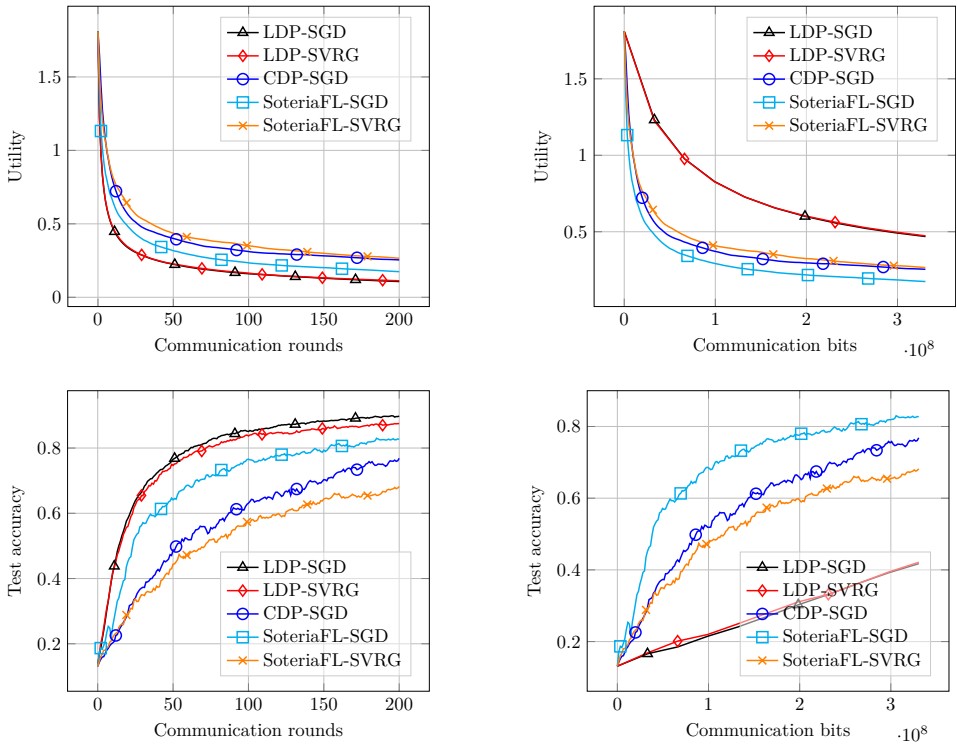

Figure 5: Shallow neural network training on the MNIST dataset under $(\epsilon, \delta)$-LDP with $\epsilon = 2$ and $\delta = 10^{-3}$. The top (resp. bottom) row is for utility (resp. test accuracy) vs. communication rounds and communication bits.

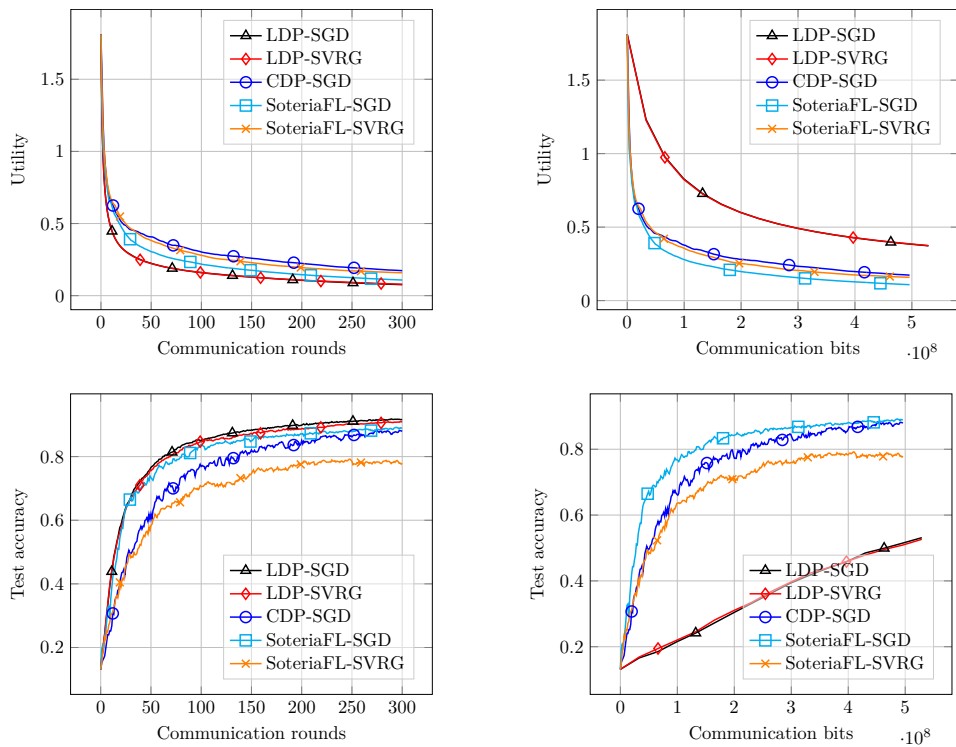

Figure 6: Shallow neural network training on the MNIST dataset under $(\epsilon, \delta)$-LDP with $\epsilon = 4$ and $\delta = 10^{-3}$. The top (resp. bottom) row is for utility (resp. test accuracy) vs. communication rounds and communication bits.

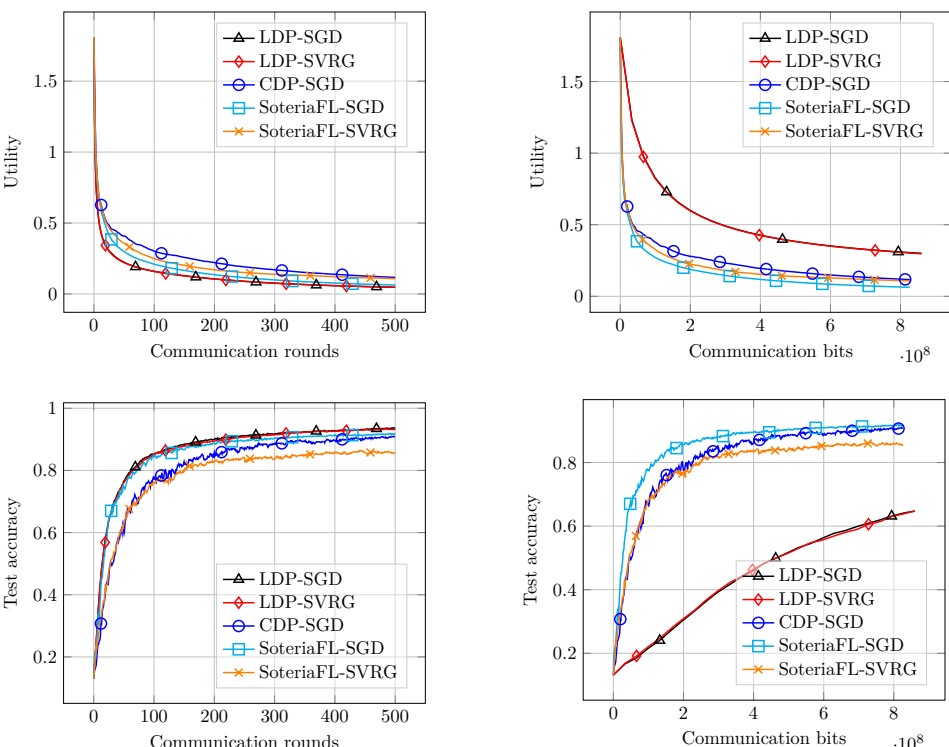

Figure 7: Shallow neural network training on the MNIST dataset under $(\epsilon, \delta)$-LDP with $\epsilon = 8$ and $\delta = 10^{-3}$. The top (resp. bottom) row is for utility (resp. test accuracy) vs. communication rounds and communication bits.

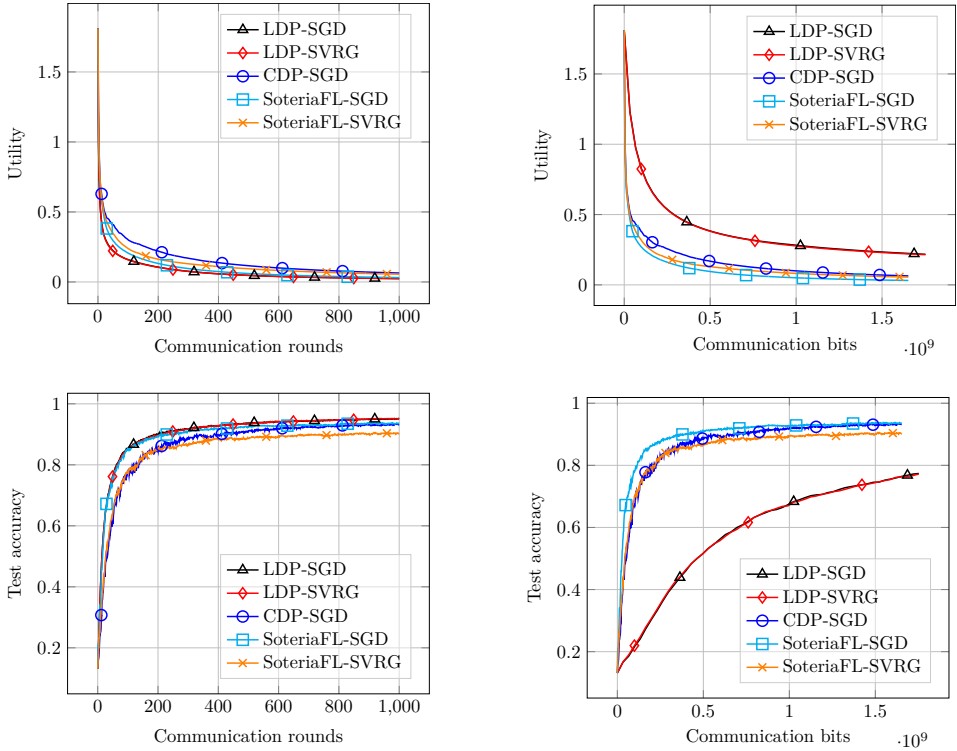

Figure 8: Shallow neural network training on the MNIST dataset under $(\epsilon, \delta)$-LDP with $\epsilon = 16$ and $\delta = 10^{-3}$. The top (resp. bottom) row is for utility (resp. test accuracy) vs. communication rounds and communication bits.

**Remark.** Note that here we not only report utility (similar as Figure 3) but also include the the test accuracy for training the neural networks (see bottom rows in Figures 4–8). The takeaways from the experimental results are similar to previous experiments on logistic regression with nonconvex regularization (Figure 3). Again, the two uncompressed algorithms (LDP-SGD and LDP-SVRG) converge faster than the three compressed algorithms (CDP-SGD, SoteriaFL-SGD, SoteriaFL-SVRG) in terms of *communication rounds* (see left columns in each figure), but the gap becomes smaller when the privacy level $\epsilon$ gets larger (i.e. less privacy guarantee). However, in terms of *communication bits* (see right columns in each figure), compressed algorithms again perform much better than the uncompressed algorithms, validating the advantage of communication compression schemes. Last but not least, shifted compression based SoteriaFL-SGD performs better than direct compression based CDP-SGD in both utility and test accuracy. However, it turns out that SoteriaFL-SVRG may perform worse than CDP-SGD for training this shallow neural network.

# B  Proof of Theorem 2

In the proof of Theorem 2, we apply a moment argument (similar to [1]) to prove the local differential privacy guarantees. Before going into the detailed proof, we first define some concepts.

**Moment generating function.**  Assume that there is a mechanism $\mathcal{M} : \mathcal{D} \to \mathcal{R}$. For neighboring datasets $D, D' \in \mathcal{D}$, a mechanism $\mathcal{M}$, auxiliary inputs aux, and an outcome $o \in \mathcal{R}$, we define the private loss at $o$ as

$$c(o; \mathcal{M}, \text{aux}, D, D') := \log \frac{\Pr\{\mathcal{M}(\text{aux}, D) = o\}}{\Pr\{\mathcal{M}(\text{aux}, D') = o\}}.$$

We also define

$$\alpha^{\mathcal{M}}(\lambda; \text{aux}, D, D') := \log \mathbb{E}_{o \sim \mathcal{M}(\text{aux}, D)} \left[ \exp\left( \lambda \cdot c(o; \mathcal{M}, \text{aux}, D, D') \right) \right]$$

and

$$\alpha_i^{\mathcal{M}}(\lambda) := \max_{\text{aux}, D, D'} \alpha^{\mathcal{M}}(\lambda; \text{aux}, D, D'),$$

where $D = (D_{-i}, D_i), D' = (D_{-i}, D'_i)$ are neighboring datasets that differ only at client $i$, $D_{-i}$ denoting all the data at clients other than client $i$. We call $\alpha^{\mathcal{M}}(\lambda; \text{aux}, D, D')$ and $\alpha_i^{\mathcal{M}}(\lambda)$ the log moment generating functions.

**Sub-mechanisms.** We assume that there are $n \times T$ sub-mechanisms $\{\mathcal{M}_i^t\}_{i \in [n], t \leq T}$ in $\mathcal{M}$, where $\mathcal{M}_i^t$ corresponds to the mechanism for client $i$ in round $t$. We further let $\mathcal{M}_i^t := \mathcal{A} \circ \overline{\mathcal{M}}_i^t$ be the composition of mechanism $\overline{\mathcal{M}}_i^t$ and the mechanism $\mathcal{A}$. Here, $\mathcal{A} : \mathcal{R} \rightarrow \mathcal{R}$ is a random mechanism that maps an outcome to another outcome, and $\overline{\mathcal{M}}_i^t$ is possibly an adaptive mechanism that takes the input of all the outputs before time $t$, i.e. $o_i^s$ for all $s < t$ and $i \in [n]$. We assume that given all the previous outcomes $o_i^s$ for $s < t$, the random mechanisms $\overline{\mathcal{M}}_i^t$ for all $i \in [n]$ are independent w.r.t. each other (this is satisfied in SoteriaFL). In SoteriaFL (Algorithm 2), $\mathcal{A}$ corresponds to the compression step, and $\overline{\mathcal{M}}_i^t$ corresponds the Gaussian perturbation.

Before proving Theorem 2, we first state the following result from [1].

**Proposition 1** (Theorem 2 in [1]). *For any $\epsilon > 0$, the mechanism $\mathcal{M}$ is $(\epsilon, \delta)$-LDP for client $i$ with $\delta = \min_\lambda \exp\left(\alpha_i^{\mathcal{M}}(\lambda) - \lambda\epsilon\right)$.*

According to Proposition 1, we know that if the log moment generating function $\alpha_i^{\mathcal{M}}(\lambda)$ is bounded, then we can show that the mechanism $\mathcal{M}$ satisfies $(\epsilon, \delta)$-LDP with some parameters $\epsilon$ and $\delta$. To prove that the log moment generating function $\alpha_i^{\mathcal{M}}(\lambda)$ is bounded, we divide it into two parts: 1) the log moment generating function $\alpha_i^{\mathcal{M}}(\lambda)$ for the whole mechanism can be bounded by the summation of the log moment generating function of all sub mechanisms $\alpha_i^{\overline{\mathcal{M}}_i^t}(\lambda)$ from $t = 1$ to $T$; and 2) the log moment generating function for each sub mechanism is bounded, i.e. $\alpha_i^{\overline{\mathcal{M}}_i^t}(\lambda)$ is bounded. To this end, we provide the following two lemmas to formalize these two parts respectively.

**Lemma 2** (Privacy for composition). *For any client $i$ and any $\lambda$, the following holds*

$$\alpha_i^{\mathcal{M}}(\lambda) \leq \sum_{t=1}^T \alpha^{\overline{\mathcal{M}}_i^t}(\lambda).$$

**Lemma 3** (Privacy for sub mechanism). *Suppose that Assumption 2 and 3 are satisfied. For any client $i$, let $\sigma_p \geq 1$ and let $\mathcal{I}_b$ be a random minibatch from local dataset $D_i = \{d_{i,j}\}_{j=1}^m$ where each data sample $d_{i,j}$ is chosen independently with probability $q = \frac{b}{m} < \frac{G_A}{16b\sigma_p}$. Then for any positive integer $\lambda \leq \frac{2b^2\sigma_p^2}{3G_A^2} \log \frac{G_A}{qb\sigma_p}$, the sub mechanism $\overline{\mathcal{M}}_i^t$ satisfies*

$$\alpha^{\overline{\mathcal{M}}_i^t}(\lambda) \leq \frac{6\lambda(\lambda+1)(G_A^2/4 + G_B^2)}{(1-q)m^2\sigma_p^2} + O\left(\frac{q^3\lambda^3}{\sigma_p^3}\right).$$

The detailed proofs for Lemmas 2 and 3 are provided in Appendix B.1 and B.2, respectively.

**Proof of Theorem 2.** Now, we are ready to prove our privacy guarantee in Theorem 2 using Proposition 1, and Lemmas 2 and 3.

*Proof of Theorem 2.* Assume for now that $\sigma_p, \lambda$ satisfy the conditions in Lemma 3, namely

$$\lambda \leq \frac{2b^2\sigma_p^2}{3G_A^2} \log \frac{G_A}{q\sigma_p b}. \tag{7}$$

By Lemmas 3 and 2, there exists some constant $\hat{c}$ such that for small enough $q$, the log moment generating function of Algorithm 2 can be bounded as follows

$$\alpha_i^{\mathcal{M}}(\lambda) \leq \hat{c}\frac{T\lambda^2(G_A^2/4 + G_B^2)}{m^2\sigma_p^2}, \qquad \forall i \in [n].$$

Combining the above bound and Proposition 1, to guarantee Algorithm 2 to be $(\epsilon, \delta)$-LDP, it suffices to establish that there exists some $\lambda$ that satisfies (7) and the following two conditions:

$$\hat{c}\frac{T\lambda^2(G_A^2/4 + G_B^2)}{m^2\sigma_p^2} \leq \frac{\lambda\epsilon}{2} \qquad \text{or equivalently} \quad \lambda \leq \frac{\epsilon m^2\sigma_p^2}{2\hat{c}T(G_A^2/4 + G_B^2)}, \tag{8}$$

$$\exp\left(-\frac{\lambda\epsilon}{2}\right) \le \delta \qquad \text{or equivalently} \quad \lambda \ge \frac{2}{\epsilon}\log\frac{1}{\delta}. \tag{9}$$

It is now easy to verify that when $\epsilon = c'q^2T$ for some constant $c'$, we can satisfy all these conditions by setting

$$\sigma_p^2 = c\frac{(G_A^2/4 + G_B^2)T\log(1/\delta)}{m^2\epsilon^2}$$

for some constant $c$. $\qquad\square$

### B.1 Proof of Lemma 2

Before embarking on the proof of Lemma 2, we begin with an observation that connects the log moment generation function with the Rényi divergence of distributions $\Pr\{\mathcal{M}(\text{aux}, D) = o\}$ and $\Pr\{\mathcal{M}(\text{aux}, D') = o\}$.

**Lemma 4.** *Denote the Rényi divergence between any two distributions $\mathcal{P}$ and $\mathcal{Q}$ with parameter $\lambda + 1$ as*

$$D_{\lambda+1}^{R\acute{e}nyi}(\mathcal{P}\|\mathcal{Q}) = \frac{1}{\lambda}\log\mathbb{E}_{\mathcal{P}}\left(\frac{d\mathcal{P}}{d\mathcal{Q}}\right)^{\lambda}.$$

*Then, the log moment generating function has the following form*

$$\alpha^{\mathcal{M}}(\lambda; aux, D, D') = \lambda D_{\lambda+1}^{R\acute{e}nyi}\left(\mathcal{M}(aux, D)\|\mathcal{M}(aux, D')\right). \tag{10}$$

*Proof of Lemma 4.* By direct computation, we have

$$
\begin{aligned}
\alpha^{\mathcal{M}}(\lambda; \text{aux}, D, D') &= \log\mathbb{E}_{o\sim\mathcal{M}(\text{aux},D)}\left[\exp\left(\lambda \cdot c(o; \mathcal{M}, \text{aux}, D, D')\right)\right] \\
&= \log\mathbb{E}_{o\sim\mathcal{M}(\text{aux},D)}\left[\exp\left(\lambda \cdot \log\frac{\Pr\{\mathcal{M}(\text{aux}, D) = o\}}{\Pr\{\mathcal{M}(\text{aux}, D') = o\}}\right)\right] \\
&= \log\mathbb{E}_{o\sim\mathcal{M}(\text{aux},D)}\left[\left(\frac{\Pr\{\mathcal{M}(\text{aux}, D) = o\}}{\Pr\{\mathcal{M}(\text{aux}, D') = o\}}\right)^{\lambda}\right] \\
&= \lambda D_{\lambda+1}^{\text{Rényi}}\left(\mathcal{M}(\text{aux}, D)\|\mathcal{M}(\text{aux}, D')\right).
\end{aligned}
$$

$\qquad\square$

We will also need the following data processing inequality for Rényi divergence.

**Lemma 5** (Data processing inequality for Rényi divergence [68]). *Let $\mathcal{P}, \mathcal{Q}$ be two distributions over $\mathcal{R}$, $\mathcal{S}: \mathcal{R} \to \mathcal{R}$ be a random mapping, and $D_{\lambda}^{R\acute{e}nyi}$ denote the Rényi Divergence, then we have*

$$D_{\lambda+1}^{R\acute{e}nyi}(\mathcal{S}(\mathcal{P})\|\mathcal{S}(\mathcal{Q})) \le D_{\lambda+1}^{R\acute{e}nyi}(\mathcal{P}\|\mathcal{Q}),$$

*where $\mathcal{S}(\mathcal{P})$ stands for the resulting distribution of applying random mapping $\mathcal{S}$ on distribution $\mathcal{P}$.*

*Proof of Lemma 2.* We divide the proof of Lemma 2 into two steps: 1) $\alpha_i^{\mathcal{M}}(\lambda) \le \sum_{t=1}^{T}\alpha^{\mathcal{M}_i^t}(\lambda)$; and 2) $\alpha^{\mathcal{M}_i^t}(\lambda) \le \alpha^{\overline{\mathcal{M}_i^t}}(\lambda)$. Combining these two steps directly leads to the declared bound, namely

$$\alpha_i^{\mathcal{M}}(\lambda) \le \sum_{t=1}^{T}\alpha^{\mathcal{M}_i^t}(\lambda) \le \sum_{t=1}^{T}\alpha^{\overline{\mathcal{M}_i^t}}(\lambda).$$

The rest of this proof is thus dedicated to establishing the two steps. For simplicity, we use $o_{1:n}^{1:T}$ to denote the outcomes $\{o_i^t\}_{i\in[n],t\in[T]}$, and $\mathcal{M}_{1:n}^{1:T}$ to denote the mechanisms $\{\mathcal{M}_i^t\}_{i\in[n],t\in[T]}$.

**Step 1: establishing** $\alpha_i^{\mathcal{M}}(\lambda) \leq \sum_{t=1}^{T} \alpha^{\mathcal{M}_i^t}(\lambda)$. For neighboring datasets $D = (D_{-i}, D_i), D' = (D_{-i}, D_i')$ that differ only on client $i$, we have

$$c(o_{1:n}^{1:T}; \mathcal{M}_{1:n}^{1:T}, o_{1:n}^{1:T-1}, D, D') = \log \frac{\Pr\{\mathcal{M}_{1:n}^{1:T}(o_{1:n}^{1:T-1}, D) = o_{1:n}^{1:T}\}}{\Pr\{\mathcal{M}_{1:n}^{1:T}(o_{1:n}^{1:T-1}, D') = o_{1:n}^{1:T}\}}$$

$$= \log \prod_{t=1}^{T} \prod_{j=1}^{n} \frac{\Pr\{\mathcal{M}_j^t(o_{1:n}^{1:T-1}, D) = o_j^t\}}{\Pr\{\mathcal{M}_j^t(o_{1:n}^{1:T-1}, D') = o_j^t\}}$$

$$= \log \prod_{t=1}^{T} \frac{\Pr\{\mathcal{M}_i^t(o_{1:n}^{1:T-1}, D) = o_i^t\}}{\Pr\{\mathcal{M}_i^t(o_{1:n}^{1:T-1}, D') = o_i^t\}}$$

$$= \sum_{t=1}^{T} \log \frac{\Pr\{\mathcal{M}_i^t(o_{1:n}^{1:T-1}, D) = o_i^t\}}{\Pr\{\mathcal{M}_i^t(o_{1:n}^{1:T-1}, D') = o_i^t\}}$$

$$= \sum_{t=1}^{T} c(o_i^t; \mathcal{M}_i^t, o_{1:n}^{1:T-1}, D, D').$$

Here, the second line comes from the fact that the mechanisms of different clients at the same round are independent, and the third line comes from the fact that for any client $j \neq i$, $D_j = D_j'$, and thus $\frac{\Pr\{\mathcal{M}_j^t(o_{1:n}^{1:T-1}, D) = o_j^t\}}{\Pr\{\mathcal{M}_j^t(o_{1:n}^{1:T-1}, D') = o_j^t\}} = 1$. Then we have

$$\mathbb{E}_{o_{1:n}^{1:T} \sim \mathcal{M}_{1:n}^{1:T}} \left[ \exp\left( \lambda c(o_{1:n}^{1:T}; \mathcal{M}_{1:n}^{1:T}, o_{1:n}^{1:T-1}, D, D') \right) \right]$$

$$= \mathbb{E}_{o_{1:n}^{1:T} \sim \mathcal{M}_{1:n}^{1:T}} \left[ \exp\left( \lambda \sum_{t=1}^{T} c(o_i^t; \mathcal{M}_i^t, o_{1:n}^{1:T-1}, D, D') \right) \right]$$

$$= \mathbb{E}_{o_{1:n}^{1:T} \sim \mathcal{M}_{1:n}^{1:T}} \left[ \prod_{t=1}^{T} \exp\left( \lambda c(o_i^t; \mathcal{M}_i^t, o_{1:n}^{1:T-1}, D, D') \right) \right]$$

$$= \prod_{t=1}^{T} \mathbb{E}_{o_{1:n}^{1:T} \sim \mathcal{M}_{1:n}^{1:T}} \left[ \exp\left( \lambda c(o_i^t; \mathcal{M}_i^t, o_{1:n}^{1:T-1}, D, D') \right) \right]$$

$$= \prod_{t=1}^{T} \exp\left( \alpha^{\mathcal{M}_i^t}(\lambda; o_{1:n}^{1:T-1}, D, D') \right)$$

$$= \exp\left( \sum_{t=1}^{T} \alpha^{\mathcal{M}_i^t}(\lambda; o_{1:n}^{1:T-1}, D, D') \right).$$

Taking logarithm on both sides and maximizing over $o_{1:n}^{1:T-1}, D, D'$, we can show that

$$\alpha_i^{\mathcal{M}}(\lambda) \leq \sum_{t=1}^{T} \alpha^{\mathcal{M}_i^t}(\lambda).$$

**Step 2: establishing** $\alpha^{\mathcal{M}_i^t}(\lambda) \leq \alpha^{\overline{\mathcal{M}}_i^t}(\lambda)$. This step follows directly from Lemma 5. Namely, for fixed $i$ and $t$, we can compute

$$\alpha^{\mathcal{M}_i^t}(\lambda; o_{1:n}^{1:T-1}, D, D') = \lambda D_{\lambda+1}^{\text{Rényi}}\left( \mathcal{M}_i^t(o_{1:n}^{1:T-1}, D) \| \mathcal{M}_i^t(o_{1:n}^{1:T-1}, D') \right)$$

$$= \lambda D_{\lambda+1}^{\text{Rényi}}\left( (\mathcal{A} \circ \overline{\mathcal{M}}_i^t)(o_{1:n}^{1:T-1}, D) \| (\mathcal{A} \circ \overline{\mathcal{M}}_i^t)(o_{1:n}^{1:T-1}, D') \right)$$

$$\leq \lambda D_{\lambda+1}^{\text{Rényi}}\left( \overline{\mathcal{M}}_i^t(o_{1:n}^{1:T-1}, D) \| \overline{\mathcal{M}}_i^t(o_{1:n}^{1:T-1}, D') \right)$$

$$= \alpha^{\overline{\mathcal{M}}_i^t}(\lambda; o_{1:n}^{1:T-1}, D, D').$$

Then, taking the maximum over $o_{1:n}^{1:T-1}, D, D'$, we have

$$\alpha^{\mathcal{M}_i^t}(\lambda) \leq \alpha^{\overline{\mathcal{M}}_i^t}(\lambda).$$

$\square$

## B.2 Proof of Lemma 3

It is worth noting that the proof does not requires $f(\boldsymbol{x}; d_{i,j})$ to be a function with respect to the data sample $d_{i,j}$ at point $\boldsymbol{x}$, it can be any function related to $d_{i,j}$, for example, $\phi_{i,j}$ in Assumption 3. Inspired by [69], we decompose the gradient estimator into two parts and bound the privacy respectively. Now we provide the detailed proofs below.

*Proof of Lemma* 3. From Assumption 3, we first write out and decouple the sub-mechanism $\overline{\mathcal{M}}_i^t$ (corresponding to the Gaussian perturbation) as

$$\frac{1}{b}\sum_{j\in\mathcal{I}_b}\varphi_{i,j}^t + \frac{1}{m}\sum_{j=1}^m\psi_{i,j}^t + \boldsymbol{\xi}_i^t = \left(\frac{1}{b}\sum_{j\in\mathcal{I}_b}\varphi_{i,j}^t + \boldsymbol{\xi}_{i,1}^t\right) + \left(\frac{1}{m}\sum_{j=1}^m\psi_{i,j}^t + \boldsymbol{\xi}_{i,2}^t\right), \quad (11)$$

where $\boldsymbol{\xi}_i^t$ is generated from $\mathcal{N}(0, \sigma_p^2\boldsymbol{I})$ and $\boldsymbol{\xi}_{i,1}^t, \boldsymbol{\xi}_{2,1}^t$ are generated from $\mathcal{N}(0, \frac{2\sigma_p^2}{3}\boldsymbol{I}), \mathcal{N}(0, \frac{\sigma_p^2}{3}\boldsymbol{I})$ independently. Now, $\overline{\mathcal{M}}_i^t$ can be viewed as a composition of two mechanisms $\overline{\mathcal{M}}_{i,1}^t$ and $\overline{\mathcal{M}}_{i,2}^t$, where $\overline{\mathcal{M}}_{i,1}^t$ denote the first term and $\overline{\mathcal{M}}_{i,2}^t$ denote the second term in the right-hand-side (RHS) of (11). From [1, Theorem 2.1], we have

$$\alpha^{\overline{\mathcal{M}}_i^t}(\lambda) \le \alpha^{\overline{\mathcal{M}}_{i,1}^t}(\lambda) + \alpha^{\overline{\mathcal{M}}_{i,2}^t}(\lambda). \quad (12)$$

For the first term of (12), according to [1, Lemma 3], we have

$$\alpha^{\overline{\mathcal{M}}_{i,1}^t}(\lambda) \le \frac{3\lambda(\lambda+1)G_A^2}{2(1-q)m^2\sigma_p^2} + O\left(\frac{q^3\lambda^3}{\sigma_p^3}\right), \quad (13)$$

for $q = \frac{b}{m} < \frac{G_A}{16\sigma_p b}$ and any positive integer $\lambda \le \frac{2b^2\sigma_p^2}{3G_A^2}\log\frac{G_A}{q\sigma_p b}$, where we set $\sigma^2$ in [1, Lemma 3] to be $\frac{2b^2\sigma_p^2}{3G_A^2}$.

For the second term of (12), according to Lemma 4 (the relationship between Rényi divergence and the moment generating function), we have

$$\alpha^{\overline{\mathcal{M}}_{i,2}^t}(\lambda) = \lambda D_{\lambda+1}^{\text{Rényi}}(\mathcal{P}\|\mathcal{Q}), \quad (14)$$

where $\mathcal{P} = \frac{1}{m}\sum_{j=1}^m\psi_{i,j}^t + \mathcal{N}(0, \frac{2\sigma_p^2}{3}\boldsymbol{I})$ and $\mathcal{Q} = \frac{1}{m}\sum_{j=1}^m(\psi_{i,j}^t)' + \mathcal{N}(0, \frac{\sigma_p^2}{3}\boldsymbol{I})$. Here, $\{\psi_{i,j}^t, \ j \in [m]\}$ contains the functions corresponding to the data in dataset $D$, and $\{(\psi_{i,j}^t)', \ j \in [m]\}$ contains the functions corresponding to the data in dataset $D'$. We note that all functions except one in $\{\psi_{i,j}^t, \ j \in [m]\}$ and $\{(\psi_{i,j}^t)', \ j \in [m]\}$ are the same, since the datasets $D$ and $D'$ only differ by one element. According to [8, Lemma 17], we have

$$\lambda D_{\lambda+1}^{\text{Rényi}}(\mathcal{P}\|\mathcal{Q}) = \frac{3\lambda(\lambda+1)\left\|\frac{1}{m}\sum_{j=1}^m\psi_{i,j}^t - \frac{1}{m}\sum_{j=1}^m(\psi_{i,j}^t)'\right\|^2}{2\sigma_p^2} \le \frac{6\lambda(\lambda+1)G_B^2}{m^2\sigma_p^2}. \quad (15)$$

The proof is finished by combining (12)–(15). □

## C  Proof of Theorem 3

We now provide the detailed proofs for our unified Theorem 3. First, according to the update rule $\boldsymbol{x}^{t+1} = \boldsymbol{x}^t - \eta_t\boldsymbol{v}^t$ (Line 9 in Algorithm 2) and the smoothness assumption (Assumption 1), we have

$$\mathbb{E}_t[f(\boldsymbol{x}^{t+1})] \le \mathbb{E}_t\left[f(\boldsymbol{x}^t) - \eta_t\langle\nabla f(\boldsymbol{x}^t), \boldsymbol{v}^t\rangle + \frac{L\eta_t^2}{2}\|\boldsymbol{v}^t\|^2\right], \quad (16)$$

where $\mathbb{E}_t$ takes the expectation conditioned on all history before round $t$. To begin, we show that $\boldsymbol{v}^t$ is unbiased as follows:

$$\mathbb{E}_t[\boldsymbol{v}^t] = \mathbb{E}_t\left[\boldsymbol{s}^t + \frac{1}{n}\sum_{i=1}^n\boldsymbol{v}_i^t\right] = \mathbb{E}_t\left[\frac{1}{n}\sum_{i=1}^n\boldsymbol{s}_i^t + \frac{1}{n}\sum_{i=1}^n\mathcal{C}_i^t(\boldsymbol{g}_i^t - \boldsymbol{s}_i^t)\right]$$

$$= \mathbb{E}_t \left[ \frac{1}{n} \sum_{i=1}^n \boldsymbol{g}_i^t \right] = \mathbb{E}_t \left[ \frac{1}{n} \sum_{i=1}^n (\tilde{\boldsymbol{g}}_i^t + \boldsymbol{\xi}_i^t) \right] \qquad (17)$$

$$= \mathbb{E}_t \left[ \frac{1}{n} \sum_{i=1}^n \tilde{\boldsymbol{g}}_i^t \right] \qquad (18)$$

$$= \mathbb{E}_t \left[ \frac{1}{n} \sum_{i=1}^n \nabla f_i(\boldsymbol{x}^t) \right] = \nabla f(\boldsymbol{x}^t), \qquad (19)$$

where (17) follows from (2), (18) holds due to $\boldsymbol{\xi}_i^t \sim \mathcal{N}(\boldsymbol{0}, \sigma_p^2 \boldsymbol{I})$, and (19) is due to $\mathbb{E}_t[\tilde{\boldsymbol{g}}_i^t] = \nabla f_i(\boldsymbol{x}^t)$ from Assumption 3.

Plugging (18) into (16), we get

$$\mathbb{E}_t[f(\boldsymbol{x}^{t+1})] \le \mathbb{E}_t \left[ f(\boldsymbol{x}^t) - \eta_t \|\nabla f(\boldsymbol{x}^t)\|^2 + \frac{L\eta_t^2}{2} \|\boldsymbol{v}^t\|^2 \right]. \qquad (20)$$

We then bound the last term $\mathbb{E}_t[\|\boldsymbol{v}^t\|^2]$ in the follow lemma, whose proof is provided in Appendix C.1.

**Lemma 6.** *Suppose that $\boldsymbol{v}^t$ is defined and computed in Algorithm 2, we have*

$$\mathbb{E}_t[\|\boldsymbol{v}^t\|^2] \le \mathbb{E}_t \left[ \frac{(1+\omega)}{n^2} \sum_{i=1}^n \|\tilde{\boldsymbol{g}}_i^t - \nabla f_i(\boldsymbol{x}^t)\|^2 + \frac{\omega}{n^2} \sum_{i=1}^n \|\nabla f_i(\boldsymbol{x}^t) - \boldsymbol{s}_i^t\|^2 \right.$$
$$\left. + \|\nabla f(\boldsymbol{x}^t)\|^2 + \frac{(1+\omega)d\sigma_p^2}{n}. \right. \qquad (21)$$

To continue, we need to bound the first two terms in the right-hand-side of (21). The first term can be controlled via (3b) of Assumption 3. Now we show that the second term will shrink in the following lemma, whose proof is provided in Appendix C.2.

**Lemma 7.** *Suppose that Assumption 1 holds and the shift $\boldsymbol{s}_i^{t+1}$ is defined and computed in Algorithm 2. Then letting $\gamma_t = \sqrt{\frac{1+2\omega}{2(1+\omega)^3}}$, we have*

$$\mathbb{E}_t \left[ \frac{1}{n} \sum_{i=1}^n \|\nabla f_i(\boldsymbol{x}^{t+1}) - \boldsymbol{s}_i^{t+1}\|^2 \right] \le \mathbb{E}_t \left[ \left( 1 - \frac{1}{2(1+\omega)} \right) \frac{1}{n} \sum_{i=1}^n \|\nabla f_i(\boldsymbol{x}^t) - \boldsymbol{s}_i^t\|^2 \right.$$
$$+ \frac{1}{(1+\omega)n} \sum_{i=1}^n \|\tilde{\boldsymbol{g}}_i^t - \nabla f_i(\boldsymbol{x}^t)\|^2$$
$$\left. + 2(1+\omega)L^2 \|\boldsymbol{x}^{t+1} - \boldsymbol{x}^t\|^2 + \frac{d\sigma_p^2}{1+\omega} \right). \qquad (22)$$

To facilitate presentation, let us introduce the short-hand notation $\mathcal{S}^t := \frac{1}{n} \sum_{i=1}^n \|\nabla f_i(\boldsymbol{x}^t) - \boldsymbol{s}_i^t\|^2$. Then we define the following potential function

$$\Phi_t := f(\boldsymbol{x}^t) - f^* + \alpha L \Delta^t + \frac{\beta}{L} \mathcal{S}^t, \qquad (23)$$

for some $\alpha \ge 0, \beta \ge 0$. With the help of Lemmas 6 and 7, we show that this potential function decreases in each round in the following lemma, whose proof is provided in Appendix C.3.

**Lemma 8.** *Under Assumptions 1 and 3, if we choose the stepsize as*

$$\eta_t \equiv \eta \le \min \left\{ \frac{1}{(1 + 2\alpha C_4 + 4\beta(1+\omega) + 2\alpha C_3/\eta^2)L}, \frac{\sqrt{\beta n}}{\sqrt{1 + 2\alpha C_4 + 4\beta(1+\omega)}(1+\omega)L} \right\},$$

*where $\alpha = \frac{3\beta C_1}{2(1+\omega)\theta L^2}$, $\forall \beta > 0$, and the shift stepsize as $\gamma_t \equiv \sqrt{\frac{1+2\omega}{2(1+\omega)^3}}$, then we have for any round $t \ge 0$,*

$$\mathbb{E}_t[\Phi_{t+1}] \le \Phi_t - \frac{\eta_t}{2} \|\nabla f(\boldsymbol{x}^t)\|^2 + \frac{3\beta}{2(1+\omega)L}(C_2 + d\sigma_p^2). \qquad (24)$$

Given Lemma 8 and Theorem 2, now we are ready to prove Theorem 3 regarding the utility and communication complexity for SoteriaFL.

*Proof of Theorem 3.* First, we sum up (24) (Lemma 8) from round $t = 0$ to $T - 1$,

$$\sum_{t=0}^{T-1} \frac{\eta_t}{2} \mathbb{E} \|\nabla f(\boldsymbol{x}^t)\|^2 \leq \Phi_0 + \frac{3\beta}{2(1+\omega)L} \left( C_2 + d\sigma_p^2 \right) T. \tag{25}$$

Then by choosing the stepsize $\eta_t$ as in Lemma 8 and the privacy variance $\sigma_p^2 = \frac{c(G_A^2/4 + G_B^2)T \log(1/\delta)}{m^2 \epsilon^2}$ according to Theorem 2, we obtain

$$\frac{1}{T} \sum_{t=0}^{T-1} \mathbb{E} \|\nabla f(\boldsymbol{x}^t)\|^2 \leq \frac{2\Phi_0}{\eta T} + \frac{3\beta}{(1+\omega)L\eta} \left( C_2 + \frac{c(G_A^2/4 + G_B^2)dT \log(1/\delta)}{m^2 \epsilon^2} \right), \tag{26}$$

and SoteriaFL (Algorithm 2) satisfies $(\epsilon, \delta)$-LDP.

Finally, the total number of communication rounds $T$ in (5) comes from the following relations in RHS of (26)

$$\frac{2\Phi_0}{\eta T} \leq \frac{3\beta}{(1+\omega)L\eta} \frac{c(G_A^2/4 + G_B^2)dT \log(1/\delta)}{m^2 \epsilon^2},$$

$$C_2 \leq \frac{c(G_A^2/4 + G_B^2)dT \log(1/\delta)}{m^2 \epsilon^2}.$$

The utility guarantee (6) directly follows from (26) by choosing $T$ as in (5). $\square$

## C.1 Proof of Lemma 6

According to the definition of $\boldsymbol{v}^t$, we have

$$\mathbb{E}_t[\|\boldsymbol{v}^t\|^2] = \mathbb{E}_t \left[ \left\| \frac{1}{n} \sum_{i=1}^n \boldsymbol{s}_i^t + \frac{1}{n} \sum_{i=1}^n \mathcal{C}_i^t(\boldsymbol{g}_i^t - \boldsymbol{s}_i^t) \right\|^2 \right]$$

$$= \mathbb{E}_t \left[ \left\| \frac{1}{n} \sum_{i=1}^n \mathcal{C}_i^t(\boldsymbol{g}_i^t - \boldsymbol{s}_i^t) - \frac{1}{n} \sum_{i=1}^n (\boldsymbol{g}_i^t - \boldsymbol{s}_i^t) + \frac{1}{n} \sum_{i=1}^n \boldsymbol{g}_i^t \right\|^2 \right]$$

$$\leq \mathbb{E}_t \left[ \frac{\omega}{n^2} \sum_{i=1}^n \|\boldsymbol{g}_i^t - \boldsymbol{s}_i^t\|^2 \right] + \mathbb{E}_t \left[ \left\| \frac{1}{n} \sum_{i=1}^n \boldsymbol{g}_i^t \right\|^2 \right], \tag{27}$$

where the last line is due to the definition of the compression operator (2). To continue, we bound each term in (27) respectively.

- For the first term, we have

$$\mathbb{E}_t \left[ \frac{\omega}{n^2} \sum_{i=1}^n \|\boldsymbol{g}_i^t - \boldsymbol{s}_i^t\|^2 \right]$$

$$= \mathbb{E}_t \left[ \frac{\omega}{n^2} \sum_{i=1}^n \|\tilde{\boldsymbol{g}}_i^t - \boldsymbol{s}_i^t + \boldsymbol{\xi}_i^t\|^2 \right]$$

$$= \mathbb{E}_t \left[ \frac{\omega}{n^2} \sum_{i=1}^n (\|\tilde{\boldsymbol{g}}_i^t - \boldsymbol{s}_i^t\|^2 + d\sigma_p^2) \right]$$

$$= \mathbb{E}_t \left[ \frac{\omega}{n^2} \sum_{i=1}^n \|\tilde{\boldsymbol{g}}_i^t - \boldsymbol{s}_i^t\|^2 \right] + \frac{\omega d\sigma_p^2}{n}$$

$$= \mathbb{E}_t \left[ \frac{\omega}{n^2} \sum_{i=1}^n \|\tilde{\boldsymbol{g}}_i^t - \nabla f_i(\boldsymbol{x}^t) + \nabla f_i(\boldsymbol{x}^t) - \boldsymbol{s}_i^t\|^2 \right] + \frac{\omega d\sigma_p^2}{n}$$

$$= \mathbb{E}_t \left[ \frac{\omega}{n^2} \sum_{i=1}^{n} \|\tilde{\boldsymbol{g}}_i^t - \nabla f_i(\boldsymbol{x}^t)\|^2 \right] + \frac{\omega}{n^2} \sum_{i=1}^{n} \|\nabla f_i(\boldsymbol{x}^t) - \boldsymbol{s}_i^t\|^2 + \frac{\omega d \sigma_p^2}{n}, \qquad (28)$$

where the last line is due to $\mathbb{E}_t[\tilde{\boldsymbol{g}}_i^t] = \nabla f_i(\boldsymbol{x}^t)$ from Assumption 3.

- Similarly, for the second term, we have

$$\mathbb{E}_t \left[ \left\| \frac{1}{n} \sum_{i=1}^{n} \boldsymbol{g}_i^t \right\|^2 \right] = \mathbb{E}_t \left[ \left\| \frac{1}{n} \sum_{i=1}^{n} (\tilde{\boldsymbol{g}}_i^t + \boldsymbol{\xi}_i^t) \right\|^2 \right]$$

$$= \mathbb{E}_t \left[ \left\| \frac{1}{n} \sum_{i=1}^{n} \tilde{\boldsymbol{g}}_i^t \right\|^2 + \frac{d\sigma_p^2}{n} \right]$$

$$= \mathbb{E}_t \left[ \left\| \frac{1}{n} \sum_{i=1}^{n} (\tilde{\boldsymbol{g}}_i^t - \nabla f_i(\boldsymbol{x}^t) + \nabla f_i(\boldsymbol{x}^t)) \right\|^2 \right] + \frac{d\sigma_p^2}{n}$$

$$= \mathbb{E}_t \left[ \frac{1}{n^2} \sum_{i=1}^{n} \|\tilde{\boldsymbol{g}}_i^t - \nabla f_i(\boldsymbol{x}^t)\|^2 \right] + \|\nabla f(\boldsymbol{x}^t)\|^2 + \frac{d\sigma_p^2}{n}. \qquad (29)$$

The proof is completed by plugging (28) and (29) into (27).

### C.2 Proof of Lemma 7

According to the shift update (Line 6 in Algorithm 2), we have

$$\mathbb{E}_t \left[ \frac{1}{n} \sum_{i=1}^{n} \|\nabla f_i(\boldsymbol{x}^{t+1}) - \boldsymbol{s}_i^{t+1}\|^2 \right]$$

$$= \mathbb{E}_t \left[ \frac{1}{n} \sum_{i=1}^{n} \left\| \nabla f_i(\boldsymbol{x}^{t+1}) - \boldsymbol{s}_i^t - \gamma_t \mathcal{C}_i^t(\boldsymbol{g}_i^t - \boldsymbol{s}_i^t) \right\|^2 \right]$$

$$= \mathbb{E}_t \left[ \frac{1}{n} \sum_{i=1}^{n} \left\| \nabla f_i(\boldsymbol{x}^{t+1}) - \nabla f_i(\boldsymbol{x}^t) + \nabla f_i(\boldsymbol{x}^t) - \boldsymbol{s}_i^t - \gamma_t \mathcal{C}_i^t(\boldsymbol{g}_i^t - \boldsymbol{s}_i^t) \right\|^2 \right]$$

$$\leq \mathbb{E}_t \left[ \frac{1}{n} \sum_{i=1}^{n} \left( (1 + \frac{1}{\beta_t}) \left\| \nabla f_i(\boldsymbol{x}^{t+1}) - \nabla f_i(\boldsymbol{x}^t) \right\|^2 + (1 + \beta_t) \left\| \nabla f_i(\boldsymbol{x}^t) - \boldsymbol{s}_i^t - \gamma_t \mathcal{C}_i^t(\boldsymbol{g}_i^t - \boldsymbol{s}_i^t) \right\|^2 \right) \right]$$
$$\tag{30}$$

$$\leq \mathbb{E}_t \left[ (1 + \frac{1}{\beta_t}) L^2 \left\| \boldsymbol{x}^{t+1} - \boldsymbol{x}^t \right\|^2 + (1 + \beta_t) \frac{1}{n} \sum_{i=1}^{n} \left\| \nabla f_i(\boldsymbol{x}^t) - \boldsymbol{s}_i^t - \gamma_t \mathcal{C}_i^t(\boldsymbol{g}_i^t - \boldsymbol{s}_i^t) \right\|^2 \right], \qquad (31)$$

where (30) uses Young's inequality with any $\beta_t > 0$ (its choice will be specified momentarily), and (31) uses Assumption 1. The second term of (31) can be further bounded as follows:

$$\mathbb{E}_t \left[ \frac{1}{n} \sum_{i=1}^{n} \left\| \nabla f_i(\boldsymbol{x}^t) - \boldsymbol{s}_i^t - \gamma_t \mathcal{C}_i^t(\boldsymbol{g}_i^t - \boldsymbol{s}_i^t) \right\|^2 \right]$$

$$= \mathbb{E}_t \left[ \frac{1}{n} \sum_{i=1}^{n} \left( (1 - 2\gamma_t) \left\| \nabla f_i(\boldsymbol{x}^t) - \boldsymbol{s}_i^t \right\|^2 + \gamma_t^2 \|\mathcal{C}_i^t(\boldsymbol{g}_i^t - \boldsymbol{s}_i^t)\|^2 \right) \right]$$

$$\overset{(2)}{\leq} \mathbb{E}_t \left[ \frac{1}{n} \sum_{i=1}^{n} \left( (1 - 2\gamma_t) \left\| \nabla f_i(\boldsymbol{x}^t) - \boldsymbol{s}_i^t \right\|^2 + \gamma_t^2 (1 + \omega) \|\boldsymbol{g}_i^t - \boldsymbol{s}_i^t\|^2 \right) \right]$$

$$= \mathbb{E}_t \left[ \frac{1}{n} \sum_{i=1}^{n} \left( (1 - 2\gamma_t + \gamma_t^2 (1 + \omega)) \left\| \nabla f_i(\boldsymbol{x}^t) - \boldsymbol{s}_i^t \right\|^2 \right. \right.$$

$$\left. \left. + \gamma_t^2 (1 + \omega) \|\tilde{\boldsymbol{g}}_i^t - \nabla f_i(\boldsymbol{x}^t)\|^2 + \gamma_t^2 (1 + \omega) d\sigma_p^2 \right) \right], \qquad (32)$$

where the first equality follows from

$$\mathbb{E}_t\left[\langle \nabla f_i(\boldsymbol{x}^t) - \boldsymbol{s}_i^t, \mathcal{C}_i^t(\boldsymbol{g}_i^t - \boldsymbol{s}_i^t)\rangle\right] = \mathbb{E}_t\left[\|\nabla f_i(\boldsymbol{x}^t) - \boldsymbol{s}_i^t\|^2\right],$$

and the last line follows from (28). The proof is completed by plugging (32) into (31) and choosing $\beta_t = \frac{1}{1+2\omega}$ and $\gamma_t = \sqrt{\frac{1+2\omega}{2(1+\omega)^3}}$.

## C.3 Proof of Lemma 8

Recalling $\mathcal{S}^t := \frac{1}{n}\sum_{i=1}^n \|\nabla f_i(\boldsymbol{x}^t) - \boldsymbol{s}_i^t\|^2$, $\mathcal{S}^t$ can be recursively bounded by Lemma 7 as

$$\mathbb{E}_t[\mathcal{S}^{t+1}] \le \mathbb{E}_t\left[\left(1 - \frac{1}{2(1+\omega)}\right)\mathcal{S}^t + \frac{1}{(1+\omega)n}\sum_{i=1}^n \|\tilde{\boldsymbol{g}}_i^t - \nabla f_i(\boldsymbol{x}^t)\|^2\right.$$
$$\left. + 2(1+\omega)L^2\|\boldsymbol{x}^{t+1} - \boldsymbol{x}^t\|^2 + \frac{d\sigma_p^2}{1+\omega}\right]. \tag{33}$$

Note that the second term can be bounded by (3b) of Assumption 3, namely

$$\mathbb{E}_t\left[\frac{1}{n}\sum_{i=1}^n \|\tilde{\boldsymbol{g}}_i^t - \nabla f_i(\boldsymbol{x}^t)\|^2\right] \le C_1\Delta^t + C_2,$$

leading to

$$\mathbb{E}_t[\mathcal{S}^{t+1}] \le \mathbb{E}_t\left[\left(1 - \frac{1}{2(1+\omega)}\right)\mathcal{S}^t + \frac{C_1\Delta^t + C_2}{(1+\omega)n} + 2(1+\omega)L^2\|\boldsymbol{x}^{t+1} - \boldsymbol{x}^t\|^2 + \frac{d\sigma_p^2}{1+\omega}\right].$$

Combined with (20), we can bound the potential function (23) as

$$\mathbb{E}_t[\Phi_{t+1}] := \mathbb{E}_t\left[f(\boldsymbol{x}^{t+1}) - f^* + \alpha L\Delta^{t+1} + \frac{\beta}{L}\mathcal{S}^{t+1}\right]$$

$$\le \mathbb{E}_t\left[f(\boldsymbol{x}^t) - f^* - \eta_t\|\nabla f(\boldsymbol{x}^t)\|^2 + \frac{L\eta_t^2}{2}\|\boldsymbol{v}^t\|^2 + \alpha L\Delta^{t+1}\right.$$
$$\left. + \frac{\beta}{L}\left(\left(1 - \frac{1}{2(1+\omega)}\right)\mathcal{S}^t + \frac{C_1\Delta^t + C_2}{1+\omega} + 2(1+\omega)L^2\|\boldsymbol{x}^{t+1} - \boldsymbol{x}^t\|^2 + \frac{d\sigma_p^2}{1+\omega}\right)\right]$$

$$\overset{(3c)}{\le} \mathbb{E}_t\left[f(\boldsymbol{x}^t) - f^* - \eta_t\|\nabla f(\boldsymbol{x}^t)\|^2 + \frac{L\eta_t^2}{2}\|\boldsymbol{v}^t\|^2\right.$$
$$+ \alpha L\left((1-\theta)\Delta^t + C_3\|\nabla f(\boldsymbol{x}^t)\|^2 + C_4\|\boldsymbol{x}^{t+1} - \boldsymbol{x}^t\|^2\right)$$
$$\left. + \frac{\beta}{L}\left(\left(1 - \frac{1}{2(1+\omega)}\right)\mathcal{S}^t + \frac{C_1\Delta^t + C_2}{1+\omega} + 2(1+\omega)L^2\|\boldsymbol{x}^{t+1} - \boldsymbol{x}^t\|^2 + \frac{d\sigma_p^2}{1+\omega}\right)\right]$$

$$= \mathbb{E}_t\left[f(\boldsymbol{x}^t) - f^* - \eta_t\|\nabla f(\boldsymbol{x}^t)\|^2 + \left(\frac{1}{2} + \alpha C_4 + 2\beta(1+\omega)\right)L\eta_t^2\|\boldsymbol{v}^t\|^2\right.$$
$$+ \alpha L\left((1-\theta)\Delta^t + C_3\|\nabla f(\boldsymbol{x}^t)\|^2\right)$$
$$\left. + \frac{\beta}{L}\left(\left(1 - \frac{1}{2(1+\omega)}\right)\mathcal{S}^t + \frac{C_1\Delta^t + C_2}{1+\omega} + \frac{d\sigma_p^2}{1+\omega}\right)\right], \tag{34}$$

where the last line follows from the update rule $\boldsymbol{x}^{t+1} = \boldsymbol{x}^t - \eta_t\boldsymbol{v}^t$ (Line 9 of Algorithm 2). To continue, we invoke Lemma 6, which gives

$$\mathbb{E}_t[\|\boldsymbol{v}^t\|^2] \le \mathbb{E}_t\left[\frac{(1+\omega)}{n^2}\sum_{i=1}^n \|\tilde{\boldsymbol{g}}_i^t - \nabla f_i(\boldsymbol{x}^t)\|^2 + \frac{\omega}{n}\mathcal{S}^t + \|\nabla f(\boldsymbol{x}^t)\|^2 + \frac{(1+\omega)d\sigma_p^2}{n}\right]$$

$$\le \mathbb{E}_t\left[\frac{(1+\omega)}{n}C_1\Delta^t + \frac{\omega}{n}\mathcal{S}^t + \|\nabla f(\boldsymbol{x}^t)\|^2 + \frac{(1+\omega)(C_2 + d\sigma_p^2)}{n}\right],$$

where the second line uses again (3b) of Assumption 3. Plugging this back into (34), we arrive at

$$\mathbb{E}_t[\Phi_{t+1}] \le f(\boldsymbol{x}^t) - f^* + \left[\alpha(1-\theta) + \frac{\beta C_1}{(1+\omega)L^2} + \left(\frac{1}{2} + \alpha C_4 + 2\beta(1+\omega)\right)\frac{(1+\omega)C_1\eta_t^2}{n}\right]L\Delta^t$$
$$+ \left[\beta\left(1 - \frac{1}{2(1+\omega)}\right) + \left(\frac{1}{2} + \alpha C_4 + 2\beta(1+\omega)\right)\frac{\omega L^2\eta_t^2}{n}\right]\frac{\mathcal{S}^t}{L}$$
$$- \left[\eta_t - \alpha L C_3 - \left(\frac{1}{2} + \alpha C_4 + 2\beta(1+\omega)\right)L\eta_t^2\right]\|\nabla f(\boldsymbol{x}^t)\|^2$$
$$+ \left[\frac{\beta}{(1+\omega)L} + \left(\frac{1}{2} + \alpha C_4 + 2\beta(1+\omega)\right)\frac{(1+\omega)L\eta_t^2}{n}\right](C_2 + d\sigma_p^2). \tag{35}$$

Now we choose the appropriate parameters satisfying

$$\alpha(1-\theta) + \frac{\beta C_1}{(1+\omega)L^2} + \left(\frac{1}{2} + \alpha C_4 + 2\beta(1+\omega)\right)\frac{(1+\omega)C_1\eta_t^2}{n} \le \alpha, \tag{36}$$

$$\beta\left(1 - \frac{1}{2(1+\omega)}\right) + \left(\frac{1}{2} + \alpha C_4 + 2\beta(1+\omega)\right)\frac{\omega L^2\eta_t^2}{n} \le \beta, \tag{37}$$

so that the RHS of (35) can lead to the potential function $\Phi_t := f(\boldsymbol{x}^t) - f^* + \alpha L\Delta^t + \frac{\beta}{L}\mathcal{S}^t$. It is not hard to verify that the following choice of $\alpha, \beta, \eta_t$ satisfy (36) and (37):

$$\alpha \ge \frac{3\beta C_1}{2(1+\omega)L^2\theta}, \qquad \forall \beta > 0, \tag{38}$$

$$\eta_t \equiv \eta \le \frac{\sqrt{\beta n}}{\sqrt{1 + 2\alpha C_4 + 4\beta(1+\omega)}(1+\omega)L}. \tag{39}$$

Note that (39) implies

$$\left(\frac{1}{2} + \alpha C_4 + 2\beta(1+\omega)\right)\frac{(1+\omega)\eta_t^2}{n} \le \frac{\beta}{2(1+\omega)L^2}. \tag{40}$$

If we further choose the stepsize

$$\eta_t \equiv \eta \le \frac{1}{(1 + 2\alpha C_4 + 4\beta(1+\omega) + 2\alpha C_3/\eta^2)L}, \tag{41}$$

then the proof is finished by combining (35)–(41) since (35) simplifies to

$$\mathbb{E}_t[\Phi_{t+1}] \le \Phi_t - \frac{\eta_t}{2}\|\nabla f(\boldsymbol{x}^t)\|^2 + \frac{3\beta}{2(1+\omega)L}(C_2 + d\sigma_p^2).$$

# D   Proof of Theorem 1

We now give the detailed proof for Theorem 1. We first show the privacy guarantee of CDP-SGD and then derive the utility guarantee.

## D.1   Privacy guarantee of CDP-SGD

**Theorem 4** (Privacy guarantee for CDP-SGD). *Suppose Assumption 2 holds. There exist constants $c'$ and $c$ so that given the sampling probability $q = b/m$ and the number of steps $T$, for any $\epsilon < c'q^2T$ and $\delta \in (0,1)$, CDP-SGD (Algorithm 1) is $(\epsilon, \delta)$-LDP if we choose*

$$\sigma_p^2 = c\frac{G^2 T \log(1/\delta)}{m^2\epsilon^2}.$$

The proof of Theorem 4 is very similar to the proof of Theorem 2. Thus here we just point out some differences between the proof of Theorem 4 and 2.

**Sub-mechanisms.** Similar to Theorem 2, we define the sub-mechanisms in the following way. We assume that there are $n \times T$ sub-mechanisms $\{\mathcal{M}_i^t\}_{i \in [n], t \leq T}$ in $\mathcal{M}$, where $\mathcal{M}_i^t$ corresponds to the mechanism for client $i$ in round $t$. We further let $\mathcal{M}_i^t := \mathcal{A} \circ \overline{\mathcal{M}}_i^t$ be the composition of mechanism $\overline{\mathcal{M}}_i^t$ and the mechanism $\mathcal{A}$. Here, $\mathcal{A} : \mathcal{R} \to \mathcal{R}$ is a random mechanism that maps an outcome to another outcome, and $\overline{\mathcal{M}}_i^t$ is possibly an adaptive mechanism that takes the input of all the outputs before time $t$, i.e. $o_i^s$ for all $s < t$ and $i \in [n]$. We assume that given all the previous outcomes $o_i^s$ for $s < t$, the random mechanisms $\overline{\mathcal{M}}_i^t$ for all $i \in [n]$ are independent w.r.t. each other (this is satisfied in CDP-SGD). In CDP-SGD (Algorithm 1), $\mathcal{A}$ corresponds to the compression operator, and $\overline{\mathcal{M}}_i^t$ corresponds the Gaussian perturbation. The difference between the sub-mechanisms for SoteriaFL and CDP-SGD is the presence of the shift. However as the shift is known to the central server, we can omit that during the analysis of privacy.

**Privacy for composition (Lemma 2).** Here CDP-SGD can use exactly the same previous Lemma 2 since the relationship between the final mechanism and the sub-mechanism does not change.

**Privacy for sub-mechanisms (Lemma 3).** The privacy guarantee for sub-mechanisms of Theorem 4 is simpler than that for Theorem 2, since we can simply apply Lemma 3 of [1] to obtain the following bound

$$\alpha^{\overline{\mathcal{M}}_i^t}(\lambda) \leq \frac{\lambda(\lambda+1)G^2}{(1-q)m^2\sigma_p^2} + O\left(\frac{q^3\lambda^3}{\sigma_p^3}\right).$$

### D.2 Utility guarantee of CDP-SGD

To prove the convergence result, we first give the following lemma providing the mean and variance of the stochastic gradient $\tilde{g}_i^t = \frac{1}{b}\sum_{j \in \mathcal{I}_b} \nabla f_{i,j}(\boldsymbol{x}^t)$ (Line 4 in Algorithm 1).

**Lemma 9** (Variance). *Under Assumption 2, for any client $i$, the stochastic gradient estimator* $\tilde{g}_i^t = \frac{1}{b}\sum_{j \in \mathcal{I}_b} \nabla f_{i,j}(\boldsymbol{x}^t)$ *is unbiased, i.e.*

$$\mathbb{E}_t\left[\frac{1}{b}\sum_{j \in \mathcal{I}_b} \nabla f_{i,j}(\boldsymbol{x}^t)\right] = \nabla f_i(\boldsymbol{x}^t),$$

*where $\mathbb{E}_t$ takes the expectation conditioned on all history before round t. Also, we have*

$$\mathbb{E}_t\left[\left\|\frac{1}{b}\sum_{j \in \mathcal{I}_b} \nabla f_{i,j}(\boldsymbol{x}^t) - \nabla f_i(\boldsymbol{x}^t)\right\|^2\right] \leq \frac{(1-q)G^2}{b}, \tag{42}$$

*where $q = b/m$.*

*Proof.* We first show that the estimator is unbiased. Define $m$ independent Bernoulli random variables $X_{i,j}$, where $\Pr\{X_{i,j} = 1\} = q = \frac{b}{m}$. Then,

$$\mathbb{E}_t\left[\frac{1}{b}\sum_{j \in \mathcal{I}_b} \nabla f_{i,j}(\boldsymbol{x}^t)\right] = \mathbb{E}_t\left[\frac{1}{b}\sum_{j=1}^{m} X_{i,j} \nabla f_{i,j}(\boldsymbol{x}^t)\right] = \frac{1}{m}\sum_{j=1}^{m} \nabla f_{i,j}(\boldsymbol{x}^t) = \nabla f_i(\boldsymbol{x}^t).$$

Moving onto the variance bound, we have

$$\mathbb{E}_t\left[\left\|\frac{1}{b}\sum_{j \in \mathcal{I}_b} \nabla f_{i,j}(\boldsymbol{x}^t) - \nabla f_i(\boldsymbol{x}^t)\right\|^2\right] = \mathbb{E}_t\left[\left\|\frac{1}{b}\sum_{j=1}^{m} X_{i,j} \nabla f_{i,j}(\boldsymbol{x}^t) - \nabla f_i(\boldsymbol{x}^t)\right\|^2\right]$$

$$= \mathbb{E}_t\left[\left\|\sum_{j=1}^{m}\left(\frac{1}{b}X_{i,j}\nabla f_{i,j}(\boldsymbol{x}^t) - \frac{1}{m}\nabla f_{i,j}(\boldsymbol{x}^t)\right)\right\|^2\right]$$

$$= \mathbb{E}_t \left[ \sum_{j=1}^m \left\| \frac{1}{b}(X_{i,j} - q)\nabla f_{i,j}(\boldsymbol{x}^t) \right\|^2 \right] \tag{43}$$

$$= \sum_{j=1}^m \frac{(1-q)q}{b^2} \left\| \nabla f_{i,j}(\boldsymbol{x}^t) \right\|^2 \le \frac{(1-q)G^2}{b},$$

where (43) comes from the fact that random variables $X_{i,j}$ are independent, and the last line follows from the variance of Bernoulli random variables as well as Assumption 2. $\square$

With the help of the above lemma, we now prove Theorem 1.

*Proof of Theorem 1.* First, from the smoothness Assumption 1, we have

$$f(\boldsymbol{x}^{t+1}) \le f(\boldsymbol{x}^t) - \eta_t \left\langle \nabla f(\boldsymbol{x}^t), \boldsymbol{v}^t \right\rangle + \frac{L\eta_t^2}{2} \left\| \boldsymbol{v}^t \right\|^2.$$

Taking the expectation on both sides of the above inequality, we have (note that we choose constant stepsize $\eta_t \equiv \eta$ for simplicity)

$$\mathbb{E}_t[f(\boldsymbol{x}^{t+1})] \le f(\boldsymbol{x}^t) - \eta\mathbb{E}_t \left\langle \nabla f(\boldsymbol{x}^t), \boldsymbol{v}^t \right\rangle + \frac{L\eta^2}{2} \mathbb{E}_t \left\| \boldsymbol{v}^t \right\|^2. \tag{44}$$

To control $\mathbb{E}_t \left\langle \nabla f(\boldsymbol{x}^t), \boldsymbol{v}^t \right\rangle$, notice that

$$\mathbb{E}_t \left\langle \nabla f(\boldsymbol{x}^t), \boldsymbol{v}^t \right\rangle = \mathbb{E}_t \left\langle \nabla f(\boldsymbol{x}^t), \frac{1}{n}\sum_{i=1}^n \boldsymbol{v}_i^t \right\rangle = \mathbb{E}_t \left\langle \nabla f(\boldsymbol{x}^t), \frac{1}{n}\sum_{i=1}^n \mathcal{C}_i^t(\boldsymbol{g}_i^t) \right\rangle$$

$$\overset{(2)}{=} \mathbb{E}_t \left\langle \nabla f(\boldsymbol{x}^t), \frac{1}{n}\sum_{i=1}^n \boldsymbol{g}_i^t \right\rangle$$

$$= \mathbb{E}_t \left\langle \nabla f(\boldsymbol{x}^t), \frac{1}{n}\sum_{i=1}^n (\tilde{\boldsymbol{g}}_i^t + \boldsymbol{\xi}_i^t) \right\rangle$$

$$= \left\| \nabla f(\boldsymbol{x}^t) \right\|^2,$$

where the last line follows from Lemma 9 as well as the independence of the added Gaussian perturbation. Next, using the definition $\boldsymbol{v}_i^t = \mathcal{C}_i^t(\boldsymbol{g}_i^t)$ and the properties of the compression operator, we compute $\mathbb{E}_t \left\| \boldsymbol{v}^t \right\|^2$ as follows,

$$\mathbb{E}_t \left\| \boldsymbol{v}^t \right\|^2 = \mathbb{E}_t \left\| \frac{1}{n}\sum_{i=1}^n \boldsymbol{v}_i^t \right\|^2$$

$$= \mathbb{E}_t \left[ \left\| \frac{1}{n}\sum_{i=1}^n (\boldsymbol{v}_i^t - \boldsymbol{g}_i^t) \right\|^2 + \left\| \frac{1}{n}\sum_{i=1}^n \boldsymbol{g}_i^t \right\|^2 \right]$$

$$\overset{(2)}{\le} \mathbb{E}_t \left[ \frac{1}{n^2}\sum_{i=1}^n \omega \left\| \boldsymbol{g}_i^t \right\|^2 + \left\| \frac{1}{n}\sum_{i=1}^n \boldsymbol{g}_i^t \right\|^2 \right]$$

$$\le \frac{1}{n^2}\sum_{i=1}^n \omega \left( \left\| \nabla f_i(\boldsymbol{x}^t) \right\|^2 + \frac{(1-q)\mathbb{E}_t \left\| \tilde{\boldsymbol{g}}_i^t - \nabla f_i(\boldsymbol{x}^t) \right\|^2}{b} + d\sigma_p^2 \right)$$

$$\quad + \left\| \nabla f(\boldsymbol{x}^t) \right\|^2 + \frac{(1-q)\mathbb{E}_t \left\| \tilde{\boldsymbol{g}}_i^t - \nabla f_i(\boldsymbol{x}^t) \right\|^2}{bn} + \frac{d\sigma_p^2}{n}$$

$$\overset{(42)}{\le} \frac{1}{n^2}\sum_{i=1}^n \omega \left( \left\| \nabla f_i(\boldsymbol{x}^t) \right\|^2 + \frac{(1-q)G^2}{b} + d\sigma_p^2 \right) + \left\| \nabla f(\boldsymbol{x}^t) \right\|^2 + \frac{(1-q)G^2}{bn} + \frac{d\sigma_p^2}{n}$$

$$\le \left\| \nabla f(\boldsymbol{x}^t) \right\|^2 + \frac{1}{n} \left( \omega G^2 + (1+\omega)\frac{(1-q)G^2}{b} + (1+\omega)d\sigma_p^2 \right), \tag{45}$$

where the last inequality (45) follows from Assumption 2.

Plugging the above two relations back to (44), we obtain

$$\mathbb{E}_t[f(\boldsymbol{x}^{t+1})] \leq f(\boldsymbol{x}^t) - \left(\eta - \frac{L\eta^2}{2}\right)\left\|\nabla f(\boldsymbol{x}^t)\right\|^2 + \frac{L\eta^2}{2n}\left(\omega G^2 + (1+\omega)\frac{(1-q)G^2}{b} + (1+\omega)d\sigma_p^2\right).$$

By choosing $\eta \leq \frac{1}{L}$, we have

$$\mathbb{E}_t[f(\boldsymbol{x}^{t+1})] \leq f(\boldsymbol{x}^t) - \frac{\eta}{2}\left\|\nabla f(\boldsymbol{x}^t)\right\|^2 + \frac{L\eta^2}{2n}\left(\omega G^2 + (1+\omega)\frac{(1-q)G^2}{b} + (1+\omega)d\sigma_p^2\right).$$

Plugging in $\sigma_p$ (Theorem 4), telescoping over the iterations $t = 1, \ldots, T$, and rearranging terms, we can prove

$$\frac{1}{T}\sum_{t=1}^{T}\mathbb{E}\left\|\nabla f(\boldsymbol{x}^t)\right\|^2$$

$$\leq \frac{2(f(\boldsymbol{x}^0) - f^*)}{\eta T} + \frac{L\eta}{n}\left[\omega G^2 + (1+\omega)\frac{(1-q)G^2}{b} + \frac{(1+\omega)cdG^2 T\log(1/\delta)}{m^2\epsilon^2}\right]$$

$$\leq \frac{2D_f}{\eta T} + \frac{L\eta}{n}\left[\frac{(1+\omega+\omega b)}{b}G^2 + \frac{(1+\omega)cdG^2 T\log(1/\delta)}{m^2\epsilon^2}\right], \tag{46}$$

where we use the notation $D_f := f(\boldsymbol{x}^0) - f^*$. We choose $T$ and $\eta$ to satisfy

$$\eta T = \frac{m\epsilon\sqrt{nD_f}}{G\sqrt{L(1+\omega)cd\log(1/\delta)}}, \quad T \geq \frac{m^2\epsilon^2}{cd\log(1/\delta)}. \tag{47}$$

According to the relation (47) and stepsize $\eta \leq \frac{1}{L}$, we set $T = \max\left\{\frac{m\epsilon\sqrt{nLD_f}}{G\sqrt{(1+\omega)cd\log(1/\delta)}}, \frac{m^2\epsilon^2}{cd\log(1/\delta)}\right\}$ and $\eta = \min\left\{\frac{1}{L}, \frac{\sqrt{nD_f cd\log(1/\delta)}}{Gm\epsilon\sqrt{(1+\omega)L}}\right\}$. Then (46) turns out as

$$\frac{1}{T}\sum_{t=1}^{T}\mathbb{E}\left\|\nabla f(\boldsymbol{x}^t)\right\|^2 \leq \frac{2D_f}{\eta T} + \frac{L\eta}{n}\left[2(1+\omega)G^2 + \frac{(1+\omega)cdG^2 T\log(1/\delta)}{m^2\epsilon^2}\right]$$

$$\overset{(47)}{\leq} \frac{2D_f}{\eta T} + \frac{L\eta}{n}\cdot\frac{3(1+\omega)cdG^2 T\log(1/\delta)}{m^2\epsilon^2}$$

$$\overset{(47)}{\leq} \frac{2G\sqrt{D_f L(1+\omega)cd\log(1/\delta)}}{m\epsilon\sqrt{n}} + \frac{3G\sqrt{D_f L(1+\omega)cd\log(1/\delta)}}{m\epsilon\sqrt{n}}$$

$$= O\left(\frac{G\sqrt{L(1+\omega)d\log(1/\delta)}}{m\epsilon\sqrt{n}}\right).$$

$\square$

# E  Proofs for Section 5

Now we provide the proofs for the proposed SoteriaFL-style algorithms. Appendix E.1 gives the proofs for Lemma 1 which shows that some classical local gradient estimators (SGD/SVRG/SAGA) satisfy our generic Assumption 3. Appendix E.2 provides the proofs for Corollaries 1–3 which instantiate Lemma 1 in the unified Theorem 3 for obtaining detailed results for the proposed SoteriaFL-style algorithms.

## E.1  Proof of Lemma 1

We shall prove each case one by one.

**The SGD estimator.** For the local SGD estimator $\tilde{\boldsymbol{g}}_i^t = \frac{1}{b}\sum_{j\in\mathcal{I}_b}\nabla f_{i,j}(\boldsymbol{x}^t)$ (Option I in Algorithm 3), we first show that it is unbiased. To facilitate analysis, for client $i$, we introduce $m$ independent Bernoulli random variables $X_{i,j}$, where $\Pr\{X_{i,j}=1\}=\frac{b}{m}$. We have

$$\mathbb{E}_t\left[\frac{1}{b}\sum_{j\in\mathcal{I}_b}\nabla f_{i,j}(\boldsymbol{x}^t)\right] = \mathbb{E}_t\left[\frac{1}{b}\sum_{j=1}^{m}X_{i,j}\nabla f_{i,j}(\boldsymbol{x}^t)\right] = \frac{1}{m}\sum_{j=1}^{m}\nabla f_{i,j}(\boldsymbol{x}^t) = \nabla f_i(\boldsymbol{x}^t).$$

Then we show that (3a)–(3c) are satisfied for some concrete parameters. For (3a), let

$$\mathcal{A}_i^t = \frac{1}{b}\sum_{j\in\mathcal{I}_b}\nabla f_{i,j}(\boldsymbol{x}^t), \quad\text{and}\quad \mathcal{B}_i^t = 0,$$

i.e., $\varphi_{i,j}^t = \nabla f_{i,j}(\boldsymbol{x}^t)$ and $\psi_{i,j}^t = 0$. Then, $G_A = G$ (Assumption 2) and $G_b = 0$. For (3b), we have

$$\mathbb{E}_t\left[\frac{1}{n}\sum_{i=1}^{n}\|\tilde{\boldsymbol{g}}_i^t - \nabla f_i(\boldsymbol{x}^t)\|^2\right] = \mathbb{E}_t\left[\frac{1}{n}\sum_{i=1}^{n}\left\|\frac{1}{b}\sum_{j\in\mathcal{I}_b}\nabla f_{i,j}(\boldsymbol{x}^t) - \nabla f_i(\boldsymbol{x}^t)\right\|^2\right]$$

$$= \mathbb{E}_t\left[\frac{1}{n}\sum_{i=1}^{n}\left\|\frac{1}{b}\sum_{j=1}^{m}X_{i,j}\nabla f_{i,j}(\boldsymbol{x}^t) - \nabla f_i(\boldsymbol{x}^t)\right\|^2\right]$$

$$= \mathbb{E}_t\left[\frac{1}{n}\sum_{i=1}^{n}\left\|\frac{1}{m}\sum_{j=1}^{m}\left(\frac{m}{b}X_{i,j}-1\right)\nabla f_{i,j}(\boldsymbol{x}^t)\right\|^2\right]$$

$$= \frac{1}{n}\sum_{i=1}^{n}\frac{m-b}{m^2 b}\sum_{j=1}^{m}\left\|\nabla f_{i,j}(\boldsymbol{x}^t)\right\|^2$$

$$\leq \frac{(m-b)G^2}{mb}, \tag{48}$$

where (48) uses Assumption 2. According to (48), we know that the SGD estimator $\tilde{\boldsymbol{g}}_i^t$ satisfies (3b) and (3c) with

$$C_1 = C_3 = C_4 = 0, \; C_2 = \frac{(m-b)G^2}{mb}, \; \theta = 1, \; \Delta^t \equiv 0.$$

**The SVRG estimator.** For the local SVRG estimator $\tilde{\boldsymbol{g}}_i^t = \frac{1}{b}\sum_{j\in\mathcal{I}_b}(\nabla f_{i,j}(\boldsymbol{x}^t) - \nabla f_{i,j}(\boldsymbol{w}^t)) + \nabla f_i(\boldsymbol{w}^t)$ (Option II in Algorithm 3), similarly we first show that it is unbiased as follows,

$$\mathbb{E}_t\left[\frac{1}{b}\sum_{j\in\mathcal{I}_b}(\nabla f_{i,j}(\boldsymbol{x}^t) - \nabla f_{i,j}(\boldsymbol{w}^t)) + \nabla f_i(\boldsymbol{w}^t)\right]$$

$$= \mathbb{E}_t\left[\frac{1}{b}\sum_{j=1}^{m}X_{i,j}(\nabla f_{i,j}(\boldsymbol{x}^t) - \nabla f_{i,j}(\boldsymbol{w}^t)) + \nabla f_i(\boldsymbol{w}^t)\right]$$

$$= \frac{1}{m}\sum_{j=1}^{m}(\nabla f_{i,j}(\boldsymbol{x}^t) - \nabla f_{i,j}(\boldsymbol{w}^t)) + \nabla f_i(\boldsymbol{w}^t)$$

$$= \nabla f_i(\boldsymbol{x}^t) - \nabla f_i(\boldsymbol{w}^t) + \nabla f_i(\boldsymbol{w}^t)$$

$$= \nabla f_i(\boldsymbol{x}^t).$$

Then we show that (3a)–(3c) are satisfied for some concrete parameters. For (3a), let

$$\mathcal{A}_i^t = \frac{1}{b}\sum_{j\in\mathcal{I}_b}(\nabla f_{i,j}(\boldsymbol{x}^t) - \nabla f_{i,j}(\boldsymbol{w}^t)), \quad\text{and}\quad \mathcal{B}_i^t = \frac{1}{m}\sum_{j=1}^{m}\nabla f_{i,j}(\boldsymbol{w}^t),$$

i.e., $\varphi_{i,j}^t = \nabla f_{i,j}(\boldsymbol{x}^t) - \nabla f_{i,j}(\boldsymbol{w}^t)$ and $\psi_{i,j}^t = \nabla f_{i,j}(\boldsymbol{w}^t)$. Then, $G_A = 2G$ and $G_b = G$ due to Assumption 2. For (3b), we have

$$\mathbb{E}_t \left[ \frac{1}{n} \sum_{i=1}^n \| \tilde{\boldsymbol{g}}_i^t - \nabla f_i(\boldsymbol{x}^t) \|^2 \right]$$

$$= \mathbb{E}_t \left[ \frac{1}{n} \sum_{i=1}^n \left\| \frac{1}{b} \sum_{j \in \mathcal{I}_b} \left( \nabla f_{i,j}(\boldsymbol{x}^t) - \nabla f_{i,j}(\boldsymbol{w}^t) \right) + \nabla f_i(\boldsymbol{w}^t) - \nabla f_i(\boldsymbol{x}^t) \right\|^2 \right]$$

$$= \mathbb{E}_t \left[ \frac{1}{n} \sum_{i=1}^n \left\| \frac{1}{b} \sum_{j=1}^m X_{i,j} \left( \nabla f_{i,j}(\boldsymbol{x}^t) - \nabla f_{i,j}(\boldsymbol{w}^t) \right) - \left( \nabla f_i(\boldsymbol{x}^t) - \nabla f_i(\boldsymbol{w}^t) \right) \right\|^2 \right]$$

$$= \mathbb{E}_t \left[ \frac{1}{n} \sum_{i=1}^n \left\| \frac{1}{m} \sum_{j=1}^m \left( \frac{m}{b} X_{i,j} - 1 \right) \left( \nabla f_{i,j}(\boldsymbol{x}^t) - \nabla f_{i,j}(\boldsymbol{w}^t) \right) \right\|^2 \right]$$

$$= \frac{1}{n} \sum_{i=1}^n \frac{m-b}{m^2 b} \sum_{j=1}^m \left\| \nabla f_{i,j}(\boldsymbol{x}^t) - \nabla f_{i,j}(\boldsymbol{w}^t) \right\|^2$$

$$\leq \frac{L^2}{b} \| \boldsymbol{x}^t - \boldsymbol{w}^t \|^2, \tag{49}$$

where (49) uses Assumption 1. According to (49), we know that the SVRG estimator $\tilde{\boldsymbol{g}}_i^t$ satisfies (3b) with

$$C_1 = \frac{L^2}{b}, \ C_2 = 0, \ \Delta^t = \| \boldsymbol{x}^t - \boldsymbol{w}^t \|^2.$$

Finally, for (3c), we have

$$\mathbb{E}_t \left[ \Delta^{t+1} \right] = \mathbb{E}_t \left[ \| \boldsymbol{x}^{t+1} - \boldsymbol{w}^{t+1} \|^2 \right]$$

$$= \mathbb{E}_t \left[ p \| \boldsymbol{x}^{t+1} - \boldsymbol{x}^t \|^2 + (1-p) \| \boldsymbol{x}^{t+1} - \boldsymbol{w}^t \|^2 \right] \tag{50}$$

$$= \mathbb{E}_t \left[ p \| \boldsymbol{x}^{t+1} - \boldsymbol{x}^t \|^2 + (1-p) \| \boldsymbol{x}^{t+1} - \boldsymbol{x}^t + \boldsymbol{x}^t - \boldsymbol{w}^t \|^2 \right]$$

$$= \mathbb{E}_t \left[ \| \boldsymbol{x}^{t+1} - \boldsymbol{x}^t \|^2 \right] + \mathbb{E}_t \left[ (1-p) \| \boldsymbol{x}^t - \boldsymbol{w}^t \|^2 + 2(1-p) \langle \boldsymbol{x}^{t+1} - \boldsymbol{x}^t, \boldsymbol{x}^t - \boldsymbol{w}^t \rangle \right]$$

$$= \mathbb{E}_t \left[ \| \boldsymbol{x}^{t+1} - \boldsymbol{x}^t \|^2 \right] + \mathbb{E}_t \left[ (1-p) \| \boldsymbol{x}^t - \boldsymbol{w}^t \|^2 + 2(1-p) \langle -\eta_t \boldsymbol{v}^t, \boldsymbol{x}^t - \boldsymbol{w}^t \rangle \right]$$

$$\overset{(19)}{=} \mathbb{E}_t \left[ \| \boldsymbol{x}^{t+1} - \boldsymbol{x}^t \|^2 \right] + \mathbb{E}_t \left[ (1-p) \| \boldsymbol{x}^t - \boldsymbol{w}^t \|^2 + 2(1-p) \langle -\eta_t \nabla f(\boldsymbol{x}^t), \boldsymbol{x}^t - \boldsymbol{w}^t \rangle \right]$$

$$\leq \mathbb{E}_t \left[ \| \boldsymbol{x}^{t+1} - \boldsymbol{x}^t \|^2 \right]$$
$$+ \mathbb{E}_t \left[ (1-p) \| \boldsymbol{x}^t - \boldsymbol{w}^t \|^2 + \frac{(1-p)p}{2} \| \boldsymbol{x}^t - \boldsymbol{w}^t \|^2 + \frac{2(1-p)\eta_t^2}{p} \| \nabla f(\boldsymbol{x}^t) \|^2 \right] \tag{51}$$

$$\leq \left( 1 - \frac{p}{2} \right) \| \boldsymbol{x}^t - \boldsymbol{w}^t \|^2 + \frac{2(1-p)\eta^2}{p} \| \nabla f(\boldsymbol{x}^t) \|^2 + \mathbb{E}_t \left[ \| \boldsymbol{x}^{t+1} - \boldsymbol{x}^t \|^2 \right] \tag{52}$$

where (50) uses the update rule of $\boldsymbol{w}^{t+1}$ (Line 7 of Algorithm 3), (51) uses Young's inequality, and the last inequality holds by choosing $\eta \geq \eta_t$. According to (52), we know that the SVRG estimator $\tilde{\boldsymbol{g}}_i^t$ satisfies (3c) with

$$\theta = \frac{p}{2}, \ C_3 = \frac{2(1-p)\eta^2}{p}, \ C_4 = 1.$$

**The SAGA estimator.** For the local SAGA estimator $\tilde{\boldsymbol{g}}_i^t = \frac{1}{b} \sum_{j \in \mathcal{I}_b} (\nabla f_{i,j}(\boldsymbol{x}^t) - \nabla f_{i,j}(\boldsymbol{w}_{i,j}^t)) + \frac{1}{m} \sum_{j=1}^m \nabla f_{i,j}(\boldsymbol{w}_{i,j}^t)$ (Option III in Algorithm 3), similarly we first show that it is unbiased as follows,

$$\mathbb{E}_t \left[ \frac{1}{b} \sum_{j \in \mathcal{I}_b} (\nabla f_{i,j}(\boldsymbol{x}^t) - \nabla f_{i,j}(\boldsymbol{w}_{i,j}^t)) + \frac{1}{m} \sum_{j=1}^m \nabla f_{i,j}(\boldsymbol{w}_{i,j}^t) \right]$$

$$= \mathbb{E}_t \left[ \frac{1}{b} \sum_{j=1}^{m} X_{i,j} (\nabla f_{i,j}(\boldsymbol{x}^t) - \nabla f_{i,j}(\boldsymbol{w}_{i,j}^t)) + \frac{1}{m} \sum_{j=1}^{m} \nabla f_{i,j}(\boldsymbol{w}_{i,j}^t) \right]$$

$$= \frac{1}{m} \sum_{j=1}^{m} (\nabla f_{i,j}(\boldsymbol{x}^t) - \nabla f_{i,j}(\boldsymbol{w}_{i,j}^t)) + \frac{1}{m} \sum_{j=1}^{m} \nabla f_{i,j}(\boldsymbol{w}_{i,j}^t)$$

$$= \frac{1}{m} \sum_{j=1}^{m} \nabla f_{i,j}(\boldsymbol{x}^t) = \nabla f_i(\boldsymbol{x}^t).$$

Then we show that (3a)–(3c) are satisfied for some concrete parameters. For (3a), let

$$\mathcal{A}_i^t = \frac{1}{b} \sum_{j \in \mathcal{I}_b} (\nabla f_{i,j}(\boldsymbol{x}^t) - \nabla f_{i,j}(\boldsymbol{w}_{i,j}^t)) \quad \text{and} \quad \mathcal{B}_i^t = \frac{1}{m} \sum_{j=1}^{m} \nabla f_{i,j}(\boldsymbol{w}_{i,j}^t),$$

i.e., $\varphi_{i,j}^t = \nabla f_{i,j}(\boldsymbol{x}^t) - \nabla f_{i,j}(\boldsymbol{w}_{i,j}^t)$ and $\psi_{i,j}^t = \nabla f_{i,j}(\boldsymbol{w}_{i,j}^t)$. Then, $G_A = 2G$ and $G_b = G$ due to Assumption 2. For (3b), we have

$$\mathbb{E}_t \left[ \frac{1}{n} \sum_{i=1}^{n} \|\tilde{\boldsymbol{g}}_i^t - \nabla f_i(\boldsymbol{x}^t)\|^2 \right]$$

$$= \mathbb{E}_t \left[ \frac{1}{n} \sum_{i=1}^{n} \left\| \frac{1}{b} \sum_{j \in \mathcal{I}_b} (\nabla f_{i,j}(\boldsymbol{x}^t) - \nabla f_{i,j}(\boldsymbol{w}_{i,j}^t)) + \frac{1}{m} \sum_{j=1}^{m} (\nabla f_{i,j}(\boldsymbol{w}_{i,j}^t) - \nabla f_{i,j}(\boldsymbol{x}^t)) \right\|^2 \right]$$

$$= \mathbb{E}_t \left[ \frac{1}{n} \sum_{i=1}^{n} \left\| \frac{1}{b} \sum_{j=1}^{m} X_{i,j} (\nabla f_{i,j}(\boldsymbol{x}^t) - \nabla f_{i,j}(\boldsymbol{w}^t)) - \frac{1}{m} \sum_{j=1}^{m} (\nabla f_{i,j}(\boldsymbol{x}^t) - \nabla f_{i,j}(\boldsymbol{w}_{i,j}^t)) \right\|^2 \right]$$

$$= \mathbb{E}_t \left[ \frac{1}{n} \sum_{i=1}^{n} \left\| \frac{1}{m} \sum_{j=1}^{m} \left( \frac{m}{b} X_{i,j} - 1 \right) (\nabla f_{i,j}(\boldsymbol{x}^t) - \nabla f_{i,j}(\boldsymbol{w}_{i,j}^t)) \right\|^2 \right]$$

$$= \frac{1}{n} \sum_{i=1}^{n} \frac{m-b}{m^2 b} \sum_{j=1}^{m} \left\| \nabla f_{i,j}(\boldsymbol{x}^t) - \nabla f_{i,j}(\boldsymbol{w}_{i,j}^t) \right\|^2$$

$$\leq \frac{L^2}{b} \frac{1}{nm} \sum_{i=1}^{n} \sum_{j=1}^{m} \|\boldsymbol{x}^t - \boldsymbol{w}_{i,j}^t\|^2, \tag{53}$$

where (53) uses Assumption 1. According to (53), we know that the SAGA estimator $\tilde{\boldsymbol{g}}_i^t$ satisfies (3b) with

$$C_1 = \frac{L^2}{b}, \quad C_2 = 0, \quad \Delta^t = \frac{1}{nm} \sum_{i=1}^{n} \sum_{j=1}^{m} \|\boldsymbol{x}^t - \boldsymbol{w}_{i,j}^t\|^2.$$

Finally, for (3c), we have

$$\mathbb{E}_t \left[ \Delta^{t+1} \right] = \mathbb{E}_t \left[ \frac{1}{nm} \sum_{i=1}^{n} \sum_{j=1}^{m} \|\boldsymbol{x}^{t+1} - \boldsymbol{w}_{i,j}^{t+1}\|^2 \right]$$

$$= \mathbb{E}_t \left[ \frac{1}{nm} \sum_{i=1}^{n} \sum_{j=1}^{m} \left( \frac{b}{m} \|\boldsymbol{x}^{t+1} - \boldsymbol{x}^t\|^2 + \left(1 - \frac{b}{m}\right) \|\boldsymbol{x}^{t+1} - \boldsymbol{w}_{i,j}^t\|^2 \right) \right] \tag{54}$$

$$= \mathbb{E}_t \left[ \frac{1}{nm} \sum_{i=1}^{n} \sum_{j=1}^{m} \left( \frac{b}{m} \|\boldsymbol{x}^{t+1} - \boldsymbol{x}^t\|^2 + \left(1 - \frac{b}{m}\right) \|\boldsymbol{x}^{t+1} - \boldsymbol{x}^t + \boldsymbol{x}^t - \boldsymbol{w}_{i,j}^t\|^2 \right) \right]$$

$$= \mathbb{E}_t \left[ \left(1 - \frac{b}{m}\right) \frac{1}{nm} \sum_{i=1}^{n} \sum_{j=1}^{m} \left( \|\boldsymbol{x}^t - \boldsymbol{w}_{i,j}^t\|^2 + 2 \left\langle \boldsymbol{x}^{t+1} - \boldsymbol{x}^t, \boldsymbol{x}^t - \boldsymbol{w}_{i,j}^t \right\rangle \right) \right]$$

$$+ \mathbb{E}_t \left[ \|\boldsymbol{x}^{t+1} - \boldsymbol{x}^t\|^2 \right]$$

$$= \mathbb{E}_t \left[ \left( 1 - \frac{b}{m} \right) \frac{1}{nm} \sum_{i=1}^n \sum_{j=1}^m \left( \|\boldsymbol{x}^t - \boldsymbol{w}_{i,j}^t\|^2 + 2 \left\langle -\eta_t \boldsymbol{v}^t, \boldsymbol{x}^t - \boldsymbol{w}_{i,j}^t \right\rangle \right) \right]$$

$$+ \mathbb{E}_t \left[ \|\boldsymbol{x}^{t+1} - \boldsymbol{x}^t\|^2 \right]$$

$$\stackrel{(19)}{=} \mathbb{E}_t \left[ \left( 1 - \frac{b}{m} \right) \frac{1}{nm} \sum_{i=1}^n \sum_{j=1}^m \left( \|\boldsymbol{x}^t - \boldsymbol{w}_{i,j}^t\|^2 + 2 \left\langle -\eta_t \nabla f(\boldsymbol{x}^t), \boldsymbol{x}^t - \boldsymbol{w}_{i,j}^t \right\rangle \right) \right]$$

$$+ \mathbb{E}_t \left[ \|\boldsymbol{x}^{t+1} - \boldsymbol{x}^t\|^2 \right]$$

$$\leq \mathbb{E}_t \left[ \left( 1 - \frac{b}{m} \right) \frac{1}{nm} \sum_{i=1}^n \sum_{j=1}^m \left( \left( 1 + \frac{b}{2m} \right) \|\boldsymbol{x}^t - \boldsymbol{w}_{i,j}^t\|^2 + \frac{2m\eta_t^2}{b} \|\nabla f(\boldsymbol{x}^t)\|^2 \right) \right]$$

$$+ \mathbb{E}_t \left[ \|\boldsymbol{x}^{t+1} - \boldsymbol{x}^t\|^2 \right] \tag{55}$$

$$\leq \left( 1 - \frac{b}{2m} \right) \frac{1}{nm} \sum_{i=1}^n \sum_{j=1}^m \|\boldsymbol{x}^t - \boldsymbol{w}_{i,j}^t\|^2 + \frac{2(m-b)\eta^2}{b} \|\nabla f(\boldsymbol{x}^t)\|^2$$

$$+ \mathbb{E}_t \left[ \|\boldsymbol{x}^{t+1} - \boldsymbol{x}^t\|^2 \right], \tag{56}$$

where (54) uses the update rule of $\boldsymbol{w}_{i,j}^{t+1}$ (Line 10 of Algorithm 3), (55) uses Young's inequality, and the last inequality holds by choosing $\eta \geq \eta_t$. According to (56), we know that the SAGA estimator $\tilde{\boldsymbol{g}}_i^t$ satisfies (3c) with

$$\theta = \frac{b}{2m}, \ C_3 = \frac{2(m-b)\eta^2}{b}, \ C_4 = 1.$$

### E.2 Proofs for SoteriaFL-style Algorithms

We provide detailed corollaries and their proofs for the proposed SoteriaFL-style algorithms (SoteriaFL-GD, SoteriaFL-SGD, SoteriaFL-SVRG, and SoteriaFL-SAGA). These corollaries are obtained by plugging their corresponding parameters given in Lemma 1 into our unified Theorem 3.

**Analysis of SoteriaFL-SGD / SoteriaFL-GD (Proof of Corollary 1).** We first show that the stepsize $\eta_t$ chosen in this corollary satisfies the conditions in Theorem 3. According to the corresponding parameters for the SGD estimator in Lemma 1

$$G_A = G, \ G_B = C_1 = C_3 = C_4 = 0, \ C_2 = \frac{(m-b)G^2}{mb}, \ \theta = 1, \ \Delta^t \equiv 0, \tag{57}$$

we have $\alpha = \frac{3\beta C_1}{2(1+\omega)\theta L^2} = 0$. Then the stepsize $\eta_t \equiv \eta$ required in Theorem 3 reads

$$\eta_t \equiv \eta \leq \min \left\{ \frac{1}{(1 + 2\alpha C_4 + 4\beta(1+\omega) + 2\alpha C_3/\eta^2)L}, \frac{\sqrt{\beta n}}{\sqrt{1 + 2\alpha C_4 + 4\beta(1+\omega)}(1+\omega)L} \right\}$$

$$= \min \left\{ \frac{1}{(1 + 4\beta(1+\omega))L}, \frac{\sqrt{\beta n}}{\sqrt{1 + 4\beta(1+\omega)}(1+\omega)L} \right\}. \tag{58}$$

Let $\tau := \frac{(1+\omega)^{3/2}}{n^{1/2}}$. If we set $\beta = \frac{\tau}{2(1+\omega)}$, then $\eta_t \equiv \eta \leq \frac{1}{(1+2\tau)L}$ satisfies (58). Then according to Theorem 3 and the parameters in (57), if we choose the shift stepsize $\gamma_t \equiv \sqrt{\frac{1+2\omega}{2(1+\omega)^3}}$, and the privacy variance $\sigma_p^2 = O\left( \frac{G^2 T \log(1/\delta)}{m^2 \epsilon^2} \right)$, SoteriaFL-SGD satisfies $(\epsilon, \delta)$-LDP and the following

$$\frac{1}{T} \sum_{t=0}^{T-1} \mathbb{E} \|\nabla f(\boldsymbol{x}^t)\|^2 \leq \frac{2\Phi_0}{\eta T} + \frac{3\beta}{(1+\omega)L\eta} \left( \frac{(m-b)G^2}{mb} + \frac{cG^2 dT \log(1/\delta)}{4m^2 \epsilon^2} \right).$$

By further choosing $T$ as

$$T = \max\left\{\frac{m\epsilon\sqrt{8(1+\omega)L\Phi_0}}{\sqrt{3\beta cdG^2\log(1/\delta)}}, \frac{4(m-b)m^2\epsilon^2}{cmbd\log(1/\delta)}\right\},\tag{59}$$

SoteriaFL has the following utility (accuracy) guarantee:

$$\frac{1}{T}\sum_{t=0}^{T-1}\mathbb{E}\|\nabla f(\boldsymbol{x}^t)\|^2 \leq O\left(\max\left\{\frac{\sqrt{\beta dG^2\log(1/\delta)}}{\eta m\epsilon\sqrt{(1+\omega)L}}, \frac{(m-b)\beta G^2}{(1+\omega)mbL\eta}\right\}\right).$$

If we further set the minibatch size $b = \min\left\{\frac{m\epsilon G\sqrt{\beta}}{\sqrt{(1+\omega)Ld\log(1/\delta)}}, m\right\}$, we have $\frac{(m-b)\beta G^2}{(1+\omega)mbL\eta} \leq$

$\frac{\sqrt{\beta dG^2\log(1/\delta)}}{\eta m\epsilon\sqrt{(1+\omega)L}}$ and thus

$$\frac{1}{T}\sum_{t=0}^{T-1}\mathbb{E}\|\nabla f(\boldsymbol{x}^t)\|^2 \leq O\left(\frac{\sqrt{\beta dG^2\log(1/\delta)}}{\eta m\epsilon\sqrt{(1+\omega)L}}\right).\tag{60}$$

Then by plugging the parameters $\beta$, $\eta$, and $b$ into (59) and (60), we obtain $T = O\left(\frac{\sqrt{nL}m\epsilon}{G\sqrt{(1+\omega)d\log(1/\delta)}}(1+\sqrt{\tau})\right)$, and $\frac{1}{T}\sum_{t=0}^{T-1}\mathbb{E}\|\nabla f(\boldsymbol{x}_t)\|^2 \leq O\left(\frac{G\sqrt{(1+\omega)Ld\log(1/\delta)}}{\sqrt{n}m\epsilon}(1+\sqrt{\tau})\right)$.

For SoteriaFL-GD in which the minibatch size $b = m$, we have $\frac{(m-b)\beta G^2}{(1+\omega)mbL\eta} = 0 \leq \frac{\sqrt{\beta dG^2\log(1/\delta)}}{\eta m\epsilon\sqrt{(1+\omega)L}}$, thus the same results hold for SoteriaFL-GD as well.

**Analysis of SoteriaFL-SVRG (Proof of Corollary 2).** We first show that the stepsize $\eta_t$ chosen in this corollary satisfies the conditions in Theorem 3. According to the corresponding parameters for the SVRG estimator in Lemma 1,

$$G_A = 2G, \; G_B = G, \; C_1 = \frac{L^2}{b}, \; C_2 = 0, \; C_3 = \frac{2(1-p)\eta^2}{p}, \; C_4 = 1, \; \theta = \frac{p}{2}, \; \Delta^t = \|\boldsymbol{x}^t - \boldsymbol{w}^t\|^2,\tag{61}$$

we have $\alpha = \frac{3\beta C_1}{2(1+\omega)\theta L^2} = \frac{3\beta}{(1+\omega)pb}$. Then the stepsize $\eta_t \equiv \eta$ required in Theorem 3 reads

$$\eta \leq \min\left\{\frac{1}{(1 + 2\alpha C_4 + 4\beta(1+\omega) + 2\alpha C_3/\eta^2)L}, \frac{\sqrt{\beta n}}{\sqrt{1 + 2\alpha C_4 + 4\beta(1+\omega)}(1+\omega)L}\right\}$$

$$= \min\left\{\frac{1}{\left(1 + \frac{6\beta}{(1+\omega)pb} + 4\beta(1+\omega) + \frac{12(1-p)\beta}{(1+\omega)p^2b}\right)L}, \frac{\sqrt{\beta n}}{\sqrt{1 + \frac{6\beta}{(1+\omega)pb} + 4\beta(1+\omega)}(1+\omega)L}\right\}.\tag{62}$$

Let $\tau := \frac{(1+\omega)^{3/2}}{n^{1/2}}$. If we set $\beta = \frac{p^{4/3}b^{2/3}(1+\omega)^2\min\{1,1/\tau^2\}}{n}$, $p^{2/3}b^{1/3} \leq 1/4$ and $p \leq 1/4$, then $\eta_t \equiv \eta \leq \frac{p^{2/3}b^{1/3}\min\{1,1/\tau\}}{2L}$ satisfies (62). Then according to Theorem 3 and the parameters in (61), if we choose the shift stepsize $\gamma_t \equiv \sqrt{\frac{1+2\omega}{2(1+\omega)^3}}$, and privacy variance $\sigma_p^2 = O\left(\frac{G^2 T\log(1/\delta)}{m^2\epsilon^2}\right)$, SoteriaFL-SVRG satisfies $(\epsilon, \delta)$-LDP and the following

$$\frac{1}{T}\sum_{t=0}^{T-1}\mathbb{E}\|\nabla f(\boldsymbol{x}^t)\|^2 \leq \frac{2\Phi_0}{\eta T} + \frac{6\beta cG^2dT\log(1/\delta)}{(1+\omega)L\eta m^2\epsilon^2}.$$

If we further choose the minibatch size $b = \frac{m^{2/3}}{4}$, the probability $p = b/m$, and the number of communication round

$$T = \frac{m\epsilon\sqrt{(1+\omega)L\Phi_0}}{\sqrt{3\beta cdG^2\log(1/\delta)}} = O\left(\frac{\sqrt{nL}m\epsilon}{G\sqrt{(1+\omega)d\log(1/\delta)}}\max\{1,\tau\}\right),$$

we obtain

$$\frac{1}{T}\sum_{t=0}^{T-1}\mathbb{E}\|\nabla f(\boldsymbol{x}^t)\|^2 \leq O\left(\frac{G\sqrt{(1+\omega)Ld\log(1/\delta)}}{\sqrt{n}m\epsilon}\right).$$

**Corollary 3** (SoteriaFL-SAGA). *Suppose that Assumptions 1 and 2 hold and we combine Theorem 3 and Lemma 1, i.e., choosing stepsize $\eta_t \equiv \eta \leq \frac{\min\{1,\sqrt{n/(1+\omega)^3}\}}{3L}$, where we set $\beta = \frac{(1+\omega)^2 \min\{1,n/(1+\omega)^3\}}{3n}$, minibatch size $b = 3m^{2/3}$, shift stepsize $\gamma_t \equiv \sqrt{\frac{1+2\omega}{2(1+\omega)^3}}$, and privacy variance $\sigma_p^2 = O\left(\frac{G^2 T \log(1/\delta)}{m^2 \epsilon^2}\right)$. If we further let the communication rounds $T = O\left(\frac{\sqrt{n}Lm\epsilon}{G\sqrt{(1+\omega)d\log(1/\delta)}} \max\{1, \tau\}\right)$, where $\tau := \frac{(1+\omega)^{3/2}}{n^{1/2}}$, then SoteriaFL-SAGA satisfies $(\epsilon, \delta)$- LDP and the following utility guarantee $\frac{1}{T}\sum_{t=0}^{T-1}\mathbb{E}\|\nabla f(\boldsymbol{x}^t)\|^2 \leq O\left(\frac{G\sqrt{(1+\omega)Ld\log(1/\delta)}}{\sqrt{n}m\epsilon}\right)$.*

**Analysis of SoteriaFL-SAGA (Proof of Corollary 3).** We first show that the stepsize $\eta_t$ chosen in this corollary satisfies the conditions in Theorem 3. According to the corresponding parameters for the SAGA estimator in Lemma 1,

$$G_A = 2G,\ G_B = G,\ C_1 = \frac{L^2}{b},\ C_2 = 0,\ C_3 = \frac{2(m-b)\eta^2}{b},\ C_4 = 1,$$

$$\theta = \frac{b}{2m},\ \Delta^t = \frac{1}{nm}\sum_{i=1}^{n}\sum_{j=1}^{m}\|\boldsymbol{x}^t - \boldsymbol{w}_{i,j}^t\|^2, \tag{63}$$

we have $\alpha = \frac{3\beta C_1}{2(1+\omega)\theta L^2} = \frac{3\beta m}{(1+\omega)b^2}$. Then the stepsize $\eta_t \equiv \eta$ required in Theorem 3 becomes

$$\eta \leq \min\left\{\frac{1}{(1 + 2\alpha C_4 + 4\beta(1+\omega) + 2\alpha C_3/\eta^2)L},\ \frac{\sqrt{\beta n}}{\sqrt{1 + 2\alpha C_4 + 4\beta(1+\omega)}(1+\omega)L}\right\}$$

$$= \min\left\{\frac{1}{\left(1 + \frac{6\beta m}{(1+\omega)b^2} + 4\beta(1+\omega) + \frac{12\beta m(m-b)}{(1+\omega)b^3}\right)L},\ \frac{\sqrt{\beta n}}{\sqrt{1 + \frac{6\beta m}{(1+\omega)b^2} + 4\beta(1+\omega)}(1+\omega)L}\right\}. \tag{64}$$

Let $\tau := \frac{(1+\omega)^{3/2}}{n^{1/2}}$. If we set $\beta = \frac{(1+\omega)^2 \min\{1, 1/\tau^2\}}{3n}$ and $b = 3m^{2/3}$, then $\eta_t \equiv \eta \leq \frac{\min\{1, 1/\tau\}}{3L}$ satisfies (64). Then according to Theorem 3 and the parameters in (63), if we choose the shift stepsize $\gamma_t \equiv \sqrt{\frac{1+2\omega}{2(1+\omega)^3}}$, and the privacy variance $\sigma_p^2 = O\left(\frac{G^2 T \log(1/\delta)}{m^2 \epsilon^2}\right)$, SoteriaFL-SAGA satisfies $(\epsilon, \delta)$- LDP and the following

$$\frac{1}{T}\sum_{t=0}^{T-1}\mathbb{E}\|\nabla f(\boldsymbol{x}^t)\|^2 \leq \frac{2\Phi_0}{\eta T} + \frac{2(1+\omega)cG^2 dT \log(1/\delta)\min\{1, 1/\tau^2\}}{nL\eta m^2 \epsilon^2}.$$

If we further choose the number of communication round

$$T = \frac{m\epsilon\sqrt{nL\Phi_0}}{\sqrt{(1+\omega)cdG^2\log(1/\delta)\min\{1, 1/\tau^2\}}} = O\left(\frac{\sqrt{n}Lm\epsilon}{G\sqrt{(1+\omega)d\log(1/\delta)}}\max\{1, \tau\}\right),$$

we obtain

$$\frac{1}{T}\sum_{t=0}^{T-1}\mathbb{E}\|\nabla f(\boldsymbol{x}^t)\|^2 \leq O\left(\frac{G\sqrt{(1+\omega)Ld\log(1/\delta)}}{\sqrt{n}m\epsilon}\right).$$