# OpenReview forum: "SoteriaFL: A Unified Framework for Private Federated Learning with Communication Compression"
_NeurIPS.cc/2022/Conference — NeurIPS 2022 Accept_

### Official Review · Reviewer_qXn9 · 2022-06-25

**Rating:** 4
**Confidence:** 2
**Soundness:** 3 good
**Presentation:** 2 fair
**Contribution:** 2 fair

**Summary:**

The paper presents methods for differentially-private stochastic gradient descent, bounding the communication complexity for several gradient descent methods in their framework.

**Questions:**

See issue above on definitions.

It really seems for your analysis you don't care about what type of compression is being used (since you're adding noise, then compressing) and just need the compressor to satisfy (2)?  If so it feels like you could streamline things -- it wasn't clear to me where "compression" was important to the results, and my impression after going over it a few times is that it really isn't, you just happen to be compressing the end vector.  (To me it's a bit strange to refer to this as a unified treatment -- it seems that compression is just a black box to you.)

See issue above on "practice".  Could you do an experiment?  Or, can you better justify why the theoretical results should be of sufficient interest?



**Strengths And Weaknesses:**

Strengths:

Appears to improve on current state-of-the-art communication complexity results.
Mathematical analysis.
Attempt to make the analysis as general as possible.

Weaknesses:

Essentially all theoretical for what is, at its heart, a problem for practice.  Some type of experiment showing its utility, comparing to other existing schemes, etc. seems called for.
Analysis appears to use straightforward methods;  could be clearer about the novelty.

Originality:  Original analysis.
Quality:  Proofs seem correct, haven't checked them in detail.
Clarity:  Adequate, some issues, examples below.
Significance:  Hard to tell  without experiments of some form.  Is this just a theoretical result?  Should people doing FL in practice care?

Issues:  It's unclear in reading the main paper what the "databases" are and what it means for them to be "neighbors" in this context.  What exactly are the "entries" in the database, what does the "mechanism" correspond to here -- what exactly is it that is being kept "private".  (If I'm doing learning on a set of images, is an entry one of the images?  What is the mechanism acting on the image?  What are possible effects of changing out an image?)

I have strong hesitations on this type of "theoretical" result for DP on a practical problem without any sort of experiment to see if the end-to-end product is actually worthwhile.  Asymptotics can cover up a lot.  What is the actual effect/cost of adding DP into a system on any sort of real problem?  (I have no sense from reading the paper.)

---

> ### Author Response · Authors · 2022-08-02
> **Response to Reviewer qXn9**
>
> Dear Reviewer XdPn,
>
> Thanks a lot for your review and thoughtful questions! Below is a point-to-point response to your questions.
>
> ## Numerical experiments
>
> Thanks for your suggestions! We have added numerical experiments in the revised version to support our theoretical findings. Please see the post to all reviewers and Appendix A (Page 15) in the revised supplementary material PDF for more details.
>
> ## Definition on LDP
>
> Here, the dataset/database corresponds to the data stored on a single client, and each entry is a data point/sample. If you are classifying a set of images, each entry is a single image.
>
> The neighbor datasets mean that two datasets $D, D'$ with the same size $|D| = |D'| = m$ differ at only one entry/data point, i.e. there exists $i\in [m]$ such that $d_i \neq d'_i$, while $d_j = d'_j$ for all $j\neq i$ and $j\in [m]$. Here, $d_j$ and $d'_j$ denote the $j$-th data point stored on $D$ and $D'$ respectively. For example, two neighbor image datasets will only differ in a single image.
>
> Generally speaking, local differential privacy (LDP) means that the server cannot (or is hard to) differentiate two neighbor datasets $D_j$ and $D_j'$ stored on client $j$ while training the model. In an image classification task, if you substitute a single image with another one to get a neighbor training dataset, then the LDP algorithms guarantee that the server cannot know if the dataset is changed or not after training on the dataset. It is also worth noting that DP is widely applied in practice, e.g., Apple utilizes DP in IOS systems.
>
> The mechanism is an "abstraction" of the protocol/procedure that the clients interact with the server. In the federated learning setting, a mechanism can be referred to the training algorithm, e.g., stochastic gradient descent (SGD). The input of the mechanism (algorithm) is the dataset (the images for the image classification task), and the output is the gradient information during all training iterations, which is all the information the server knows (and needs to know) to replicate the training procedure.
>
> However, vanilla SGD algorithm is not guaranteed to be differentially private, since if you change a data point, it reflects on the gradient information and thus the server can potentially know if you change the data or not ([63] "Deep leakage from gradients." by Zhu et al., NeurIPS'19.). Our SoteriaFL framework is a locally differentially-private training algorithm/procedure that protects LDP with communication compression. If you have a set of images for an image classification task, SoteriaFL framework will add noise on the "gradient" information to "fool" the server, and then the server "cannot tell if you change an image or not".

---

> ### Author Response · Authors · 2022-08-02
> **Continue to response to Reviewer qXn9**
>
> ## Unified framework and the role of compression operators
>
> First, we would like to highlight that communication cost is one of the main bottlenecks in federated learning (FL). For private FL, it is extremely important to holistically consider privacy-utility-communication trade-offs (see Section 1.1).
> As you mentioned, in our work, compression is performed after adding privacy perturbation noise. Although the compression does not affect the privacy analysis, it does affect utility and communication results and analyses. To our best knowledge, previous communication-efficient DP algorithms only work for specific or very limited compressors [15,60], or they need to design sophisticated procedures (especially the perturbation procedure) to make communication efficient and protect privacy [51,11,12,32]. Our framework is very simple (also allows more advanced shifted compression), and can work with any compression operators that satisfy Definition 1. This is one aspect of our unified framework which accommodates a general family of unbiased compression operators.
>
> Second, our SoteriaFL framework also supports the clients to use many local optimization methods as long as it satisfies the generic Assumption 3. In our paper, we currently support and provide detailed analysis for the clients to use SGD/SVRG/SAGA (see Algorithm 3 and Lemma 1), and our framework can possibly support more local optimization methods (potentially even optimization methods designed in the future). By using our framework, the clients do not need to re-calculate how to make their algorithms private from scratch. Instead, our unified theorems offer plug-in privacy-utility-communication results when they change from one local optimization algorithm to another. In our numerical experiments, it is shown that different local optimization algorithms may perform differently on different tasks, and thus supporting different local optimizers is quite useful. This is the second aspect of our unified framework which accommodates a general family of local gradient estimators.
>
> Finally, it is also worth noting that unifying different local methods in private FL is a highly non-trivial contribution, since different local optimization algorithms not only have different update steps (mechanisms), but also lead to different privacy/utility/communication analyses. Thus our unified analysis requires a high level of abstraction that captures many interesting local optimization methods, as well as more general and novel proof approaches.

---

> ### Author Response · Authors · 2022-08-06
> **Did we address your concerns successfully?**
>
> Dear Reviewer qXn9,
>
> As the discussion deadline is quickly approaching, we are eager to find out if our detailed response and new experiments successfully address your concerns. If yes, we will appreciate if you could raise the score to reflect the change, and we will be happy to address any remaining concerns otherwise. Looking forward to hearing from you!
>
> Respectfully,
> Authors

---

### Official Review · Reviewer_XdPn · 2022-07-06

**Rating:** 7
**Confidence:** 3
**Soundness:** 4 excellent
**Presentation:** 4 excellent
**Contribution:** 3 good

**Summary:**

This paper studies FL framework that both take into account compression to reduce the communication cost, and Local Differential-Privacy. The paper first introduces a baseline method CDP-SGD, which is basically DP-SGD combined with a compression of the model updates, and compute its utility for a number of communication rounds and a (epsilon, delta) DP bound.

The paper then introduced another class of framework (SoteriaFL). The main idea of SoteriaFL is to introduce a local shift s_i at each clients, that is track across the rounds. The gradient update $g_i$ of each client is shifted by $s_i$ before being compressed and sent to the central server. The server also keep track of a global shift, and add this global shift back to the sum of the model updates received during the aggregation part of the FL algorithm. SoteriaFL can be used with various local gradient estimators. The paper consider three such gradient estimators: SGD, SVRG and SAGA. For each of them,  the paper computes the utility and the communication complexity of SoteriaFL under (epsilon,delta)-Local Differential Privacy constraints.

**Questions:**

- Could you elaborate more on the assumptions 1 and 2, and explain in which FL cases you think there are satisfied?


**Limitations:**

I see to main limitations of this work:

- There are no numerical experiment, even on mock data to support and illustrates the claim made by the authors. The theoretical results are $O( \cdot. )$ meaning that all the constant have disappear. Therefore it would be nice to see these results in a numerical examples in order to better understand the order of magnitudes involved.

- The authors do not put enough in context the necessity of such a framework. It would be better to have an example of a real-world FL use-case, in order to put order of magnitude in the different parameters (number of clients, dataset etc.) and show explicitly that SoteriaFL could improved real-world FL project.

I don't think that these two limitations prevent from publication, as this work is a theoretical work and the focus of the paper is on convergence/compression rate.

**Strengths And Weaknesses:**

# Originality
The method presented in the paper seems new, and the authors made a specific effort to cite related work and explain how their work differ from existing work. In particular they point out that their CDP-SGD algorithm can subsume other previously published works when specific hyperparameter are chosen.


# Quality

The submission seems technically sound, and the mathematical proof of the claims made are provided. I did not check all the proofs of the papers (and the appendices). But the quality of the writing and the clarity of the presentation indicate a meticulous work from the authors.


# Clarity
The paper is extremely clear and well-written. I just have the following two remarks:

- Algorithm 1, line 3: $I_b$ is not defined. It is defined later at line 204, but it would be important to define it here properly. In particular, it is important for the reader to understand that this step is only a gradient estimation, and there is no local model updates made as in FedAvg.

- Algorithm2: Tjhe initialization of $s_i^{(t=0)}$ and $s^{(t=0)}$ is not specified. Could you specify it?



# Significance
Computing theoretical convergence rates and complexity bounds of newly designed FL framework is important in order to provide more theoretical foundations to the FL research fields. Therefore the line of research followed by this paper is worthy of publication. However it's significance is slightly reduced by the fact that:

- The authors do not justify when/whether the assumptions 1 and 2 are true
- There is not simulations, even on mock data, to illustrate and support there findings.

---

> ### Author Response · Authors · 2022-08-02
> **Response to Reviewer XdPn**
>
> Dear  Reviewer XdPn,
>
> Thanks for your positive comments and thoughtful questions! Below is a point-to-point response to your questions.
>
> ## Numerical experiments
> Thanks for your suggestions! We have added numerical experiments in the revised version to support our theoretical findings. Please see the post to all reviewers and Appendix A (Page 15) in the revised supplementary material PDF for more details.
>
> ## Justification of Assumptions 1 and 2
>
> The smoothness constant $L$ (Assumption 1) depends on the data and the problem, and it is a standard assumption for establishing convergence results. For example, in linear regression $\min_x f(x) = \sum_i (x^\top a_i-b_i)^2$, it follows that $L=\|\sum_i a_ia_i^\top\|$ satisfies Assumption 1. However, $L$ is typically unknown (or hard to compute) in closed form for nonconvex problems such as neural networks. Note that $L$ is usually related to the learning rate/stepsize of the algorithms (e.g. $1/L$ for standard gradient descent), and thus, people usually tune the learning rate to achieve the best performance. In our numerical experiments (Appendix A), we also tune the learning rate properly for every algorithm. One strength of the theory lies in providing some intuitions for convergence: algorithms that support a larger learning rate in theory often can use a larger learning rate in practice as well, and thus converge/perform better.
>
>
> For the bounded gradient Assumption 2, it is required due to the privacy guarantee. Similar to the smoothness parameter $L$, it is also not easy to obtain an upper bound on the gradient norm (i.e. bounded gradient parameter $G$) in real-world applications. Thus in practice, people usually apply the gradient clipping technique to clip the gradient (i.e. $\text{clip}_G(g)= \min (1, \frac{G}{\| g \|} ) \cdot g$, then $\|\text{clip}_G(g)\|\leq G$ satisfies Assumption 2). This technique is extensively used in the numerical experiments of many differential privacy literature, where researchers assume Assumption 2 for theoretical analysis, and use  gradient clipping in experiments [57,60,15,43]. We also use gradient clipping and compute the privacy perturbation variance given the gradient clipping parameter $G$ in our experiments.
>
> ## Questions on clarity
>
> Thanks for your suggestion! We have added the definition of $I_b$ in Algorithm 1 in the revised version.
>
> The shift $s_i^{(0)}$ can be initialized as $0$. We have added this specification in Algorithm 2 in the revised version. In our numerical experiments, we also use the initialization $s_i^{(0)} = 0$ for SoteriaFL.
>
> ## Regarding the real-world FL use-case
>
> It is an excellent suggestion to develop real-world FL use-cases for SoteriaFL. While the current paper focuses on theoretical analysis and laying down the groundwork, it is of great interest to further develop real-world use-cases in the next step. We imagine SoteriaFL will be highly relevant in bandwidth-limited internet-of-things (IoT) applications such as smart home, where it necessitates the use of communication compression due to bandwidth limitation between the edge devices with the server, and the use of privacy-preserving algorithms to respect the privacy of personalized devices (things).

---

> > ### Comment · Reviewer_XdPn · 2022-08-08
> > **Answer to authors**
> >
> > Dear authors,
> >
> > I appreciate the fact that the authors added numerical experiments in their work that support their conclusions and their main assumptions. I still score the paper as 7/10. Even though, this is not my main domain of expertise, I think that this paper should be published.

---

> > > ### Author Response · Authors · 2022-08-08
> > > **Re: Answer to authors**
> > >
> > > Dear Reviewer XdPn,
> > >
> > > Thank you so much for reading the response and keeping the support of our paper!

---

> ### Author Response · Authors · 2022-08-06
> **A kind reminder**
>
> Dear Reviewer XdPn,
>
> We hope that our newly added experiments and the detailed justification of Assumptions 1 and 2 successfully address your concerns. If yes, we will appreciate if you could keep and consider increasing your support, and we will be happy to answer any other questions if you have. Looking forward to hearing from you!
>
> Respectfully, Authors

---

### Official Review · Reviewer_YLXD · 2022-07-10

**Rating:** 8
**Confidence:** 4
**Soundness:** 4 excellent
**Presentation:** 3 good
**Contribution:** 4 excellent

**Summary:**

The paper introduces a private and communication-efficient framework for federated learning. For communication efficiency, the authors use compression, and for privacy, the authors use local differential privacy. The authors first start with a simple algorithm that applies compression directly to DP-SGD, and expand it to the SoteriaFL framrwork, which supports a number of of local gradient estimators and the state-of-the-art shifted compression scheme. The authors back up their claims with theoretical and mathematical rigor.

**Questions:**

The only question I have from authors, is that, are the current theoretical analyses enough to back the arguments? How about some field experiments with actual devices and real measurements? Will something like that increase the confidence of the claims of the paper?



**Limitations:**

It would be nice if the authors upfront and honest discuss the limitations of the paper. I believe it adds value and increases the trust from the readers

(disclaimer: I should mention that I did not check the correctness of the theorems and proofs)

**Strengths And Weaknesses:**

Strengths:
- The paper is well-written and it's very easy to follow
- The paper has a smooth transition from a simple algorithm that applies compression directly to DP-SGD, to a more advanced private and communication-efficient algorithm. I personally found this transition smooth and nice.
- The paper is backed by a lot of theory and it is technically sound

Weakness:
- I am not a true expert in DP papers, while I read a lot of them, but I think this paper could benefit from some numerical analysis and experiments.

---

> ### Author Response · Authors · 2022-08-02
> **Response to Reviewer YLXD**
>
> Dear Reviewer YLXD,
>
> Thanks a lot for your positive feedback and suggestions! We have added some numerical experiments to corroborate our theoretical findings in the revised PDF. Please see the post to all reviewers and Appendix A (Page 15) in the revised supplementary material PDF for more details.
>
> Regarding possible limitations, as SoteriaFL considers the general family of unbiased compression operators, it might not be applied directly if the compression operator is biased, such as the top-$k$ compression. Therefore, it is of interest to further extend the framework of SoteriaFL to handle biased compression operators.
>
> Besides, we only conducted numerical simulations, not field experiments with actual devices and real measurements as you mentioned due to resource limitation. It is certainly meaningful to test our framework on real systems in the future.

---

> ### Author Response · Authors · 2022-08-06
> **A kind reminder**
>
> Dear Reviewer YLXD,
>
> We hope that our newly added experiments answer your only question well.
> We will appreciate if you could keep and consider increasing your support, and we will be happy to answer any other questions if you have. Looking forward to hearing from you!
>
> Respectfully, Authors

---

> > ### Comment · Reviewer_YLXD · 2022-08-08
> > **Thank you for addressing my comments**
> >
> > Dear authors,
> >
> > Thank you for addressing my comments. I am happy to see that a whole section of experiments were added. Appreciate adding that. This adds a big value to the paper.
> >
> > Regarding the limitation, thank you for pointing them out. Make sure you include them in the paper.
> >
> > I increased my score for this paper.

---

> > > ### Author Response · Authors · 2022-08-08
> > > **Glad we have addressed your concerns**
> > >
> > > Dear Reviewer YLXD,
> > >
> > > Thank you so much for reading the response and increasing the support of our paper! We will update the final version as discussed.
> > >
> > > Authors

---

### Author Response · Authors · 2022-08-02
**To all reviewers**

Dear reviewers,

Thanks for your valuable comments, suggestions, and questions. In this post, we address the major concern pointed out by all reviewers.

## [Major] Numerical experiments

All reviewers point out that our paper can be benefited from additional numerical experiments. Thanks for this valuable comment, and we have added some numerical experiments in the revised appendix (Appendix A, pages 15--21) to corroborate our theoretical findings. Below we briefly summarize our numerical setup and results.

Setup:
We conduct two experiments: logistic regression with non-convex regularization on the a9a dataset, and shallow neural network training on the MNIST dataset.

In each experiment, we compare 5 algorithms:
+ LDP-SGD: DP algorithm without communication compression [1,43]
+ LDP-SVRG: DP algorithm without communication compression [43]
+ CDP-SGD: Algorithm 1.
+ SoteriaFL-SGD: Algorithm 3, Option I.
+ SoteriaFL-SVRG: Algorithm 3, Option II.

We use the random sparsification compressor in both experiments, where the compression ratio is 20 (i.e., transmit 5% of the coordinates after compression). In other words, the communication cost of 1 communication round of uncompressed algorithms (LDP-SGD and LDP-SVRG) equals to that of 20 rounds of compressed algorithms (CDP-SGD, SoteriaFL-SGD, and SoteriaFL-SVRG).

For all algorithms, we optimize the learning rate (stepsize) to make sure that the algorithm under evaluation attains the best performance.

The main takeaways of logistic regression with non-convex regularization are summarized as follows.
+ The uncompressed algorithms (LDP-SGD, LDP-SVRG) converge faster than the compressed algorithms in terms of communication rounds.
+ In terms of the communication bits, all compressed algorithms perform better than the uncompressed algorithms.
+ SoteriaFL performs better than CDP-SGD, in both the utility and the training loss.
+ SoteriaFL-SVRG performs better than SoteriaFL-SGD.

As for shallow neural network training, most of the findings in logistic regression with non-convex regularization remain the same.
+ The uncompressed algorithms (LDP-SGD, LDP-SVRG) converge faster than the compressed algorithms in terms of communication rounds.
+ In terms of communication bits, all compressed algorithms perform better than the uncompressed algorithms.
+ SoteriaFL-SGD always performs better than CDP-SGD. However, SoteriaFL-SVRG may perform worse than CDP-SGD in this shallow neural network training task.

In sum, we believe the numerical experiments confirm the superiority or competitiveness of SoteriaFL to achieve better communication complexity without sacrificing privacy nor utility than other private federated learning algorithms without communication compression.

---

### Meta-Review · Area_Chair_LQad · 2022-08-30

**Recommendation:** Accept
**Confidence:** Certain

**Metareview:**

The paper focuses on two important challenges in federated learning: communication efficiency and privacy of the users' local data. There has been a large body of work studying each of these challenges individually, but there are not many prior works which consider both of these challenges simultaneously. The paper proposes a unified framework that enhances the communication efficiency of private federated learning with communication compression. The paper provides two types of results: (i) It theoretically analyses the case in which a compression method is applied directly to differentially-private stochastic gradient descent; Using this analysis, the authors then explain some limitations of this approach. (ii) The authors then propose a framework, called SoteriaFL, which accommodates a general family of local gradient estimators (which includes SGD, SAGA, SVRG, etc) and a shifted compression scheme (which is one of the well-known methods for compression). SoteriaFL is then analyzed theoretically and performance trade-offs in terms of privacy, utility, and communication complexity are obtained. The benefits of SoteriaFL are then shown (in terms of communication complexity and less privacy, utility loss) compared to other methods. Table I provides a good summary.

All in all, the reviewers found the theoretical results novel and interesting. The reviewers had a number of concerns most of which were addressed by the authors response and through the discussions. I thank the authors for their responses and also for adding experimental results (which was mentioned by all the reviewers). Two of the reviewers are very positive about the paper. The third reviewer has also read the author responses and mentioned through private (email) communication that they'd be willing to increase their score if the experimental results are reasonable. All in all, based on the reviewers' feedback (and my own assessment) I think this paper provides an array of novel contributions in the field of FL.


**Award:**

No

---

### Decision · Program_Chairs · 2022-09-14

Accept